# Common and Novel Markers for Measuring Inflammation and Oxidative Stress Ex Vivo in Research and Clinical Practice—Which to Use Regarding Disease Outcomes?

**DOI:** 10.3390/antiox10030414

**Published:** 2021-03-09

**Authors:** Alain Menzel, Hanen Samouda, Francois Dohet, Suva Loap, Mohammed S. Ellulu, Torsten Bohn

**Affiliations:** 1Laboratoires Réunis, 38, Rue Hiehl, L-6131 Junglinster, Luxembourg; amenzel@pt.lu (A.M.); dnalliance@gmail.com (F.D.); 2Nutrition and Health Research Group, Department of Population Health, Luxembourg Institute of Health, 1 A-B, Rue Thomas Edison, L-1445 Strassen, Luxembourg; hanene.samouda@lih.lu; 3Clinic Cryo Esthetic, 11 Rue Éblé, 75007 Paris, France; suvaloap@gmail.com; 4Department of Clinical Nutrition, Faculty of Applied Medical Sciences, Al-Azhar University of Gaza (AUG), Gaza City 00970, Palestine; mohdsubhilulu@gmail.com

**Keywords:** transcription factors, immune system, diet, chronic diseases, reactive oxygen species, superoxide, dietary inflammatory index, acute phase proteins, cell counting

## Abstract

Many chronic conditions such as cancer, chronic obstructive pulmonary disease, type-2 diabetes, obesity, peripheral/coronary artery disease and auto-immune diseases are associated with low-grade inflammation. Closely related to inflammation is oxidative stress (OS), which can be either causal or secondary to inflammation. While a low level of OS is physiological, chronically increased OS is deleterious. Therefore, valid biomarkers of these signalling pathways may enable detection and following progression of OS/inflammation as well as to evaluate treatment efficacy. Such biomarkers should be stable and obtainable through non-invasive methods and their determination should be affordable and easy. The most frequently used inflammatory markers include acute-phase proteins, essentially CRP, serum amyloid A, fibrinogen and procalcitonin, and cytokines, predominantly TNFα, interleukins 1β, 6, 8, 10 and 12 and their receptors and IFNγ. Some cytokines appear to be disease-specific. Conversely, OS—being ubiquitous—and its biomarkers appear less disease or tissue-specific. These include lipid peroxidation products, e.g., F2-isoprostanes and malondialdehyde, DNA breakdown products (e.g., 8-OH-dG), protein adducts (e.g., carbonylated proteins), or antioxidant status. More novel markers include also –omics related ones, as well as non-invasive, questionnaire-based measures, such as the dietary inflammatory-index (DII), but their link to biological responses may be variable. Nevertheless, many of these markers have been clearly related to a number of diseases. However, their use in clinical practice is often limited, due to lacking analytical or clinical validation, or technical challenges. In this review, we strive to highlight frequently employed and useful markers of inflammation-related OS, including novel promising markers.

## 1. Introduction

“Inflammation” describes a sequence of reactions of the immune system in response to often, but not always, harmful stimuli. These can be infections (bacteria, viruses, parasites), but also injuries, physical and chemical phenomena (burns, radiation, etc.), tissue injury (necrosis), auto-immune or hypersensitivity reactions, i.e., allergies [1]. The “aim” of the inflammatory response is firstly to eliminate the causal stimulus, and ultimately, to form temporary replacement tissue as part of the healing process. However, many chronic diseases are characterized by low-grade chronic inflammation, accompanied by oxidative stress (OS). Due to its implication in many pathological events and related mortality, inflammation has been termed the “secret killer” in some non-scientific literature. Prominent examples of health complications associated with inflammation and OS include obesity [2], metabolic syndrome (MetS), [3], type-2 diabetes (T2D) [4], atherosclerosis and other cardiovascular diseases (CVD) [5], chronic kidney disease as well as auto-immune diseases including inflammatory bowel diseases (IBD) such as Crohn’s disease (CD) and ulcerative colitis (UC) [6]. Additional acute phases of these diseases are characterized by flare-ups or bursts of inflammation, i.e., acute inflammation. These are characterized by much higher concentrations of inflammatory and OS markers, including various cytokines and reactive oxygen species (ROS). Examples also include acute infections, a present example being COVID-19 [7]. However, inflammation can result in cellular damage, tissue erosion, cancer, organ failure, shock and death [8].

Thus, inflammation becomes deleterious when it is insufficient to contain the primary insult, when it extends and becomes systemic, or when it lasts for extended periods of time and becomes chronic. Thus, inflammation is a normal and meaningful response of the human body to injury or infection. In case of the latter, the innate immune-system (as opposed to the adaptive one) reacts to invading microorganisms by activating phagocytes such as macrophages, dendritic cells, neutrophils or monocytes, releasing cytokines and activating the complement system. Coagulation factors are also activated. This enhances the immune response by promoting stimulation and differentiation of immune cells, including natural killer (NK) cells, macrophages and mast cells, leading to their polarization into various subtypes, as well as stimulating their migration [9]. These also trigger the activation of enzymes that generate ROS, such as NADPH oxidases, to damage invaders [7]. A certain level of ROS generation is physiological and required for optimal cellular functioning [10]. ROS are also produced in the mitochondria, as a result of respiratory aerobic decarboxylation. That a certain amount of ROS is needed to fight infection is supported by abnormal high infection rates in subjects incapable of producing superoxide radicals (O_2_^•−^) [11].

To diagnose inflammation and OS (Figure 1), and to monitor disease progression and treatment, markers of inflammation and OS can be measured in the clinical context. The purpose is to diagnose both symptomatic and pre-symptomatic diseases. Monitoring their levels also contributes to the follow-up of the response to the treatment. As such they also may have a prognostic value. A biomarker should preferably be measurable through non- or minimal invasive procedures. Body fluids and in particular urine and blood are therefore preferred media. They should also have a reasonable half-life, be stable and not affected by collection or storage conditions. Their assay also should be simple, robust and affordable. These latter aspects are often a limitation in clinical facilities, as many more sophisticated instruments such as LC-MS-MS are not employed in clinical practice. However, most important is not only the methodological and analytical validation of a marker, but also their clinical validation, i.e., a proven clinical relevance and significance based on clinical trials with clearly defined endpoints. These trials should allow determining reference values in healthy populations with regard to gender and age, distribution and dispersion of these values (standard deviation), as well as defining cut-off values to diseased conditions.

One can distinguish two main families of biomarkers of inflammation, which are currently in use, though rarely in a clinical background. Firstly, the first-line-of-defence circulating cytokines such as interferons (IF) and interleukins (IL), e.g., IL-1β, IL-6, IL-8, IFγ and TNF-α, which can be produced by immune cells including macrophages, B-and T-lymphocytes, mast cells, and endothelial cells. These have also frequently been measured for disease prediction, such as for chronic heart failure [12]. Several of them are not routinely measured as they are costly and susceptible to storage and freeze-thaw cycling [13]. Second in line are the acute phase proteins (APPs), some of which can inhibit microbial growth. These are typically formed early during inflammation and secreted by the liver, such as C-reactive protein (CRP, often measured in clinical facilities), haptoglobin, or serum amyloid a (SAA). These APPs have shown in several meta-analyses to be well correlated with clinical outcomes and disease occurrence, including obesity [14] and cardiovascular events [15], among others.

Regarding OS, ROS may be measured directly, such as by electrospin resonance (EPR) [16], but this is usually not realized in clinical practice or even research due to equipment restrictions and sample instability. Rather, the resulting oxidized compounds, mostly proteins, lipids and RNA/DNA are measured. These include malondialdehyde (MDA), shown to correlate with several chronic diseases such as Parkinson’s [17]. Measuring DNA/RNA degradation products such as by 8-hydroxy-2′-deoxyguanosine (8-OH-dG) can be an alternative, shown for example to be elevated in CVD according to a recent meta-analysis [18]. This marker is also fairly stable during long-term storage [19], compared to many other markers.

An alternative to assessing ROS is to measure counterbalancing antioxidants. Such “total antioxidant capacity (TAC) assays” include e.g., ABTS (2,2′-azino-bis(3-ethylbenzothiazoline-6-sulfonic acid)), the ferric reducing antioxidant power assay (FRAP) [20], or—also intracellularly—ratios of oxidized vs. reduced glutathione. However, often either a water- or lipo-soluble fraction/extract is investigated, not both, thus not capturing the entire scope of antioxidants. Also measuring antioxidant enzymes such as catalase (CAT), superoxide-dismutase (SOD) or glutathione peroxidase (GPx) can be carried out [21]. As for antioxidant activity, these constitute a rather indirect measure of OS. An interesting and more physiological approach is the ex vivo measurement of (e.g., copper) induced lipoprotein oxidation, to assess their stability against peroxidation [22], or oxidized LDL (oxLDL), a marker correlating with atherosclerosis [23].

As situated physiologically further upstream of cytokines or antioxidant enzymes, transcription factors are more centrally involved in the relay of inflammation and OS. Thus, they may constitute an interesting and perhaps more integrative marker. Candidates include nuclear factor kappa B (NF-κB) and nuclear factor erythroid 2 related factor 2 (Nrf2), which can be measured by antibodies. However, as especially their translocation to the nucleus should be assessed, rather than total cellular concentration, a cellular sample such as white-blood cells is needed, and isolation of the nuclear fraction is advised [24]. NF-κB expression has been (positively) associated in a meta-analysis with worse tumor outcomes [25], similar as in a meta-analysis for Nrf2 (negative association) and breast cancer [26].

“Composite” or “index” markers combining a variety of individual markers may provide more insights into OS and inflammation compared to individual markers. For OS alone a large number of such indices was proposed, often including the thiol or glutathione ratio [27]. For inflammation, a score composed of white blood cell (WBCs), erythrocyte sedimentation rate, CRP, soluble TNF-α and β receptors was successfully applied to insulin resistance conditions [28]. While such indices may be more accurate, with perhaps higher disease-predictability, they require multiple-marker assessment. Another novel tool, -omics techniques, could also offer interesting insights, but is even more costly to measure, and more challenging to interpret. Finally, also questionnaires have been proposed to offer a non-invasive, cheap alternative tool for assessing inflammatory status. However, they are often not validated, or rather address a specific disease, such as IBD [29]. A nowadays frequently used index related to dietary patterns, is the dietary inflammatory index (DII), which has shown to correlate well with inflammatory markers, including CRP and IL-6 [30]. However, this marker considers only dietary aspects and relies on rather time-consuming detailed dietary assessment such as by food-frequency questionnaires (FFQs).

In this review, we aim to provide an overview of markers of inflammation and OS which are prominently utilized in research and in clinical practice and especially *ex vivo*, trying to highlight their usefulness and limitations, also emphasizing novel promising candidates which may aid in the prediction of inflammation and OS related diseases.

## 2. Origin and Physiological Aspects of Oxidative Stress

### 2.1. Oxidative Damage, Products and Transcription Factors

#### 2.1.1. Damage to Lipid Molecules

Free radicals produced during OS can react with susceptible molecules in their vicinity within cells or biological fluids. Especially susceptible are lipids, proteins, and nucleic acids, but also carbohydrates can be damaged too [31].

Lipids play an important role in organisms, for instance as components of cellular membranes, important for cellular compartmentalization, and their oxidative damage may have pathological consequences. Mechanistically, lipid oxidation or rather peroxidation (LPO) proceeds as a radical chain reaction and is divided into the phases of initiation, propagation and termination. Chain reaction is initiated by ROS that abstract a hydrogen atom, typically from a methylene group (initiation, [32]), producing fatty acyl radicals. During the fast propagation or chain reaction, these acyl radicals react further with oxygen to produce peroxyl radicals, which are able to abstract additional hydrogen atoms, producing further alkyl radicals. OH^•^ radicals can start chain reactions with all fatty acids, whereas O_2_^−^ reacts especially with activated fatty acids [33]. Chain termination occurs through reaction of 2 radicals producing a non-radical or by reaction with quenching molecules (antioxidants), forming stable radicals. During LPO and the associated fragmentation, a wide variety of saturated and unsaturated reactive molecules with adverse physiological properties including genotoxicity, such as alkanes, aldehydes, ketones and furans can be formed, some of which may constitute markers of LPO [34].

For example, MDA, a reactive dialdehyde, is formed in vivo from polyunsaturated fatty acids (PUFAs), especially arachidonic acid and longer molecules, reacting with a peroxyl radical and O_2_, followed by cyclization and fragmentation. This happens through OS conditions in lipid membranes. However, MDA can also result from enzyme-catalyzed conversion of arachidonic acid to thromboxane. Another potential marker class of OS are F2-isoprostanes. These are non-enzymatic (without COX-1/2 involvement) reaction products of lipid oxidation. They are formed, mostly within cell membranes, out of fatty acids, predominantly arachidonic acid (but also out of eicosapentaenoic and docosahexaenoic acid), are released by the aid of phospholipases and are similar in structure to prostaglandins [35]. The different types are named and characterized according to their cyclopentane ring, with the most predominant class being the F2-isoprostanes, the earliest discovered isoprostanes. Arachidonic acid can react with free radicals to a variety of F2-isoprostanes, belonging to the 5, 8, 12 and 15 series, each composed of 16 stereoisomers.

However, PCOOHs are assumed to be the major lipid oxidation products. The majority of phospholipids in human cells are phosphatidylcholine (40–50%), and their oxidation is likely to trigger further cascades of inflammation [36]. They are associated with all cell membranes and may constitute a good marker of OS. It appears that PUFAs in the sn-2 position are most prone to oxidation [37].

#### 2.1.2. Damage to Proteins

With respect to protein damage, free radicals produced during OS can modify polypeptide chains and generate carbonylated proteins (PCs), i.e., carrying carbonyl groups (C=O). Carbonylation of proteins is considered an important hallmark of OS. It is also considered a non-reversible post-translational modification, resulting in a protein with an aldehyde, ketone or lactame moity, as reviewed by Federova et al. [38]. However, other advanced oxidation protein products (AOPPs) can be formed, often by chlorinated oxidants such as hypochloric acid [39]. Often, aldehydes and ketones, i.e., reactive carbonyl species formed during OS such as MDA react with amino acids to form adducts [40]. However, direct oxidation of proteins can also occur. Federova et al. have also highlighted the main origins of protein-bound carbonyls: These may be formed either via metal-catalyzed oxidation (side chains of amino acids Pro, Lys, Thr and Arg), direct oxidation (Trp), lipid-peroxidation (Cys, Lys, His) and also by glycation/glycoxidation (Lys, Arg). Other reactions such as hydroxylations have also been reported. From the point of view of biomarkers for OS, the modifications of aromatic amino acids (phenylalanine, tyrosine) are especially of interest. Phenylalanine is sensitive to hydroxyl radicals, which cause its conversion to *ortho*- and *meta*-tyrosine [41], which can have deleterious effects on cells. In this respect, metal-catalyzed protein oxidation is a common pathway of PC generation, in which hydroxyl-radicals are produced via the Fenton reaction (involving iron or copper), which may constitute the most common PC formation in living cells [42]. The most so produced products include glutamic semialdehyde formed from arginine and proline, and aminoadipic semialdehyde from lysine [43].

Protein nitration, such as of tyrosine residues, is another important modification, and is an aspect of nitrative stress. Since this beyond the scope of this review, the reader is referred to additional review articles [44]. In brief, following hydrogen abstraction from the phenol ring by radical mechanisms, the resulting tyrosyl radical can react with peroxynitrite (ONOOH) to form 3-nitrotyrosine [45]. In Vivo, nitration of extracellular proteins is probably due to leakage of peroxynitrite from activated macrophages, endothelial or other cells during inflammation. Enhanced levels of nitroproteins have been detected in various fluids and tissues, including plasma, liver cells, platelets, in individuals with various pathological conditions [46,47]. Circulating nitroproteins provide thus an excellent opportunity for investigating nitrative stress. This is important, as nitro-OS precedes OS and its effects are regarded as more reversible, constituting a potential therapeutic target. Unfortunately, although there are many biomarkers for evaluating OS, few have been proposed for nitrative stress. Therefore, nitrated proteins and PCs are assumed to constitute the most abundant measures of OS and their effects on proteins.

#### 2.1.3. Damage to DNA/RNA

Damage of DNA/RNA may result in mutation and genetic malfunction, causing cell damage and eventually cancer. A variety of stressors including free radicals have been reported to accelerate breakdown and increase DNA [48] and RNA [49] damage. More specifically, double stranded DNA in the nucleus may be broken down, or free bases of the cellular and especially mitochondrial deoxynucleoside triphosphate pool [50]. Oxidative DNA damage can result from reactions with purine and pyrimidine bases, deoxyribose or phosphodiesters. Especially the hydroxyl radical has been reported to form adducts with the double bonds of DNA bases or abstract hydrogen atoms from deoxyribose residues [51]. Another important reaction is the reaction of HO^•^ with purines by adding on the double bond at position 7,8, forming 8-oxo-7,8-dihydrodeoxyguanosine (8-OH-dG) out of 8-OH-dGTP and removal of the diphosphate rest, followed by further digestion of 8-OH-dGMP or DNA excision repair mechanism [52]. This compound appears to be more abundant than 8-oxo-7,8-dihydro-deoxyadenosine (8-OH-dA) [53]. Thus, 8-OH-dG is a frequently investigated indicator of endogenous oxidative DNA damage. Other breakdown products include 8-oxo-2′-deoxyguanosine (formed from 8-OH-dG), 8-hydroxyguanine (from DNA or RNA) and 8-hydroxyguanosine (from RNA).

#### 2.1.4. Main Transcription Factors Involved

In addition to its more direct actions, ROS can regulate the expression of many downstream genes [54] involved in inflammation and carcinogenesis [55], via influencing the transcription factors Nrf2 and NF-κB (Figure 2). The Nrf2 pathway is activated by OS, increasing the expression of enzymes participating in the antioxidant defense system [56], as well as phase II liver detoxification enzymes [57], in order to restore redox balance. Nrf2 has been considered a master regulator of OS. It is typically bound to its cytosolic inhibitor Keap1. Keap1 is rich in cysteine residues, acting as a sensor constantly monitoring OS levels in the cytosol [58]. Due to its cross-talk with NF-κB, it also plays an important role in inflammation [59], as Keap1 degrades the kinase of the inhibitor of NF-κB, thus preventing the activation of the latter. In addition, COX-2 also stimulates Nrf2, again limiting NF-kB activity, and NF-κB can also associate with CBP (CREB binding protein), a co-activator of Nrf2, thus being in direct competition for it. When OS is increased, Keap1 is activated, dissociates from Nrf2, which then migrates into the nucleus, activating the antioxidant response element (ARE). ARE then up-regulates a variety of antioxidant enzymes and detoxifying proteins [60]. This Keap1-Nrf2 pathway regulates >600 genes involved in cell protection and antioxidant defense, including GSH and SOD. Nrf2 also promotes anti-inflammatory prostaglandins and enzymes, fostering tissue healing and repair.

### 2.2. Antioxidant System

#### 2.2.1. Antioxidant Enzymes

Though not constituting direct markers of OS, complementary insights into OS homeostasis can be obtained via measuring enzymes involved in ROS removal, with a low activity possibly indicating an imbalance. A certain time-delay following OS insult is expected but may follow within minutes to hours of the stimuli. When HepG2 liver cells were exposed in vitro to diethyl maleate and *tert*-butylhydroquinone, maximum expression of Nrf2 was seen after 2 h [61]. Major enzymes associated with the quenching of ROS are SOD, CAT, and GPx. However, a low activity may also indicate a normal reaction due to the absence of ROS. A weakness of measuring antioxidant enzymes is thus their time-dependency of concentrations and that no guidelines for interpreting results exist, i.e., no concentrations which can clearly be associated with a healthy or unhealthy state, impeding its use in clinical studies.

The major function of SOD is to convert the superoxide radical anion O_2_^•−^ to H_2_O_2_ [62]. Thus, SOD has been termed a first line of defense against superoxide radicals. Hydrogen peroxide is then further reduced to water by CAT, GPx, or peroxiredoxins, also producing oxygen, but no further radicals. CAT activity has been reviewed to be high wherever higher concentrations of H_2_O_2_ are generated. Regarding body tissues, high activity is found in the liver, red blood cells, followed by kidney and adipose tissue, with lower activity in other tissues [63]. While CAT appears to exist to be present in peroxisomes, GPx and peroxiredoxins exist in different cellular locations, though depending on their different isoforms (see Section 4.5).

#### 2.2.2. Antioxidants

In addition to the endogenous enzymes involved in removing ROS, there are a number of endogenous antioxidants, including GSH, uric acid, creatinine, bilirubin, and several proteins, especially albumin [64]. GSH is a sulfur-containing tripeptide formed in the liver and is a water-soluble cellular antioxidant and an enzyme co-factor. It is present in reduced (GSH) and oxidized form (GSSG), the ratio of which may be a good marker of the redox status within the cell. Under resting conditions, this is about 100:1, [65]. GSH is considered a crucial element of cellular metabolism, required for instance for the regeneration of vitamin C/E, removal of mercury from cells, participating in phase II enzymatic reactions, in addition to singlet oxygen and radical quenching [66]. Together with GPx, it acts as an antioxidant, whereby GSH itself is oxidized. GSSG is then typically reduced back to GSH by glutathione-reductase [67].

On the extracellular side, uric acid, bilirubin and albumin contribute significantly to antioxidant capacity [68]. Though the antioxidant capacity on a molar base seems higher for uric acid, albumin appears to contribute more to the TAC (as much as 70%, owing to the presence of thiol groups [69]), due to its higher concentration (approx. 300–700 µM). Antioxidants derived from the diet include mainly vitamin C and E, carotenoids, and polyphenols, with plasma concentrations of ca. 20–50 µM [70], 30 µM [71], 1–2.5 µM [72], and around 2 µM [73], respectively. However, following biodistribution, their accumulation in the body differs, according to e.g., their lipophilicity, with carotenoids and vitamin E accumulating within cellular membranes, while phenolics and vitamin C being present rather in the cytosol.

## 3. Origin and Propagation of Inflammation

### 3.1. Immune System and Cellular Responses

While inflammatory reactions respond to a pathogenic stimuli and condition, they act in close conjunction with the immune system, e.g., in order to protect from pathogens. Both the innate and the adaptive immune system work closely together as individual components of the immune response and are mutually dependent [74]. The innate immune response is triggered as soon as an antigen contact happens. It can be further categorized into cellular (leukocytes, granulocytes, neutrophils, basophils, eosinophils, macrophages and natural killer cells (lymphocytes) and humoral components (complement system, chemokines/cytokines and APPs).

The most abundant WBCs include neutrophils, lymphocytes, monocytes, eosinophils, and basophils. Neutrophils are the most predominant class of leukocytes and the first cells recruited to sites of inflammation, interacting with various cytokines, modulating innate and adaptive immune response [75]. They can engulf bacteria by phygocytosis, but a more novel mechanism discovered is formation of neutrophil extracellular traps (NETs) for larger invaders, a process termed NETosis. Lymphocytes include a more diverse class of cells found predominantly in the lymphatic system, including natural killer cells, B-cells, and T-cells, thus playing a role in both adaptive and innate immunity [9]. Monocytes are the largest WBCs and can differentiate into dendritic cells, macrophages, and myeloid cells. As part of the innate immune system, they play important roles in fighting invaders. They may also be an independent predictor of mortality in the elderly [76]. Eosinophils, when activated, move into the affected tissue and secrete inflammatory mediators, aiding to eliminate foreign organisms [77]. Eosinophils also aid in activating mast cells, further fostering inflammation. Eosinophil numbers change due to a variety of infections, including viral and bacterial ones [78,79], but they are also implemented in allergic diseases. Basophils are further part of the innate immune system, quickly reacting to foreign organisms and substances, especially via T helper type 2 immune responses. They become activated when coming into contact with foreign molecules, IgE or specific signals from other cells [80]. Upon the entry of foreign entities such as bacteria or viruses, neutrophils are released from the blood stream, followed by monocytes, which transform into macrophages to engulf the invader by phagocytosis within lysosomes. Lysosomes of the macrophages will then degrade bacteria, and the debris is removed via the lymphatic system, especially the lymph nodes [81].

For the further uptake and clearance of the debris two types of resolvins, E-and D-resolvins are essential [82,83]. These are mainly derived from eicosapentaenoic acid (EPA) and docosahexaenoic acid (DHA) respectively, ω-3 fatty acids present especially in fish oil. Without sufficient resolvins, debris removal does not function properly, emphasizing the importance of ω-3 fatty acids as anti-inflammatory agents [84]. These resolvins have a variety of additional functions, such as suppressing NF-κB activation [85]. In pathological situations where macrophages cannot digest the bacterium or virus, the macrophage will burst, releasing lysosomal content, which may then further destroy tissue, starting a vicious circle, as more neutrophils are attracted and more inflammation is created.

Also related to the processing of dietary originating fatty acids into inflammatory messengers are COX-1 and 2. COX-2 is an inducible enzyme (unlike the constitutively active COX-1), transforming arachidonic acid into prostanoids, a group of mostly pro-inflammatory mediators including thromboxanes, prostaglandins and eicosanoids. Many nonsteroidal anti-inflammatory drugs such as aspirin or ibuprofen act by blocking COX-2, which is stimulated by many cytokines and growth factors [86]. Thromboxanes are rather implicated in blood pressure regulation as vasoconstrictors and foster platelet aggregation. Prostaglandins are vasodilators, inhibiting platelet aggregation and inflammatory processes. Several prostaglandins with various properties exist, the most abundant being PGE2 [87]. It relays its activity via binding to G-protein coupled (transmembrane) receptors (GPCR), and has been reported to activate mast cells and differentiation of T1 helper cells, as well as IL-22 production and proliferation of Th-17 cells, playing an important role in bacterial and fungal defence [88]

Differing from the innate system, the adaptive (specific) immune system develops in the course of our lives through daily contact with antigens. It forms antibodies through plasma B cells. Cellular components of the adaptive immune system include leukocytes, antigen-presenting cells, T-lymphocytes e.g., T-helper cells, regulatory T-cells, cytotoxic T-cells, B-lymphocytes, plasma cells and memory cells. Humoral components are especially antibodies, secreted by B-lymphocytes [89]. The humoral, antibody-mediated immune response takes place in three main phases [90]; in a first step the pathogen is recognized and tagged by macrophages as “foreign”. A conversion process then enables the macrophage to attract T-helper cells, which in suit will make contact with the B-lymphocytes, activating them. The latter then forms the antibody-producing plasma cells, producing specific antibodies against the pathogen. These finally bind to the pathogens and “tag” them for other cells of the immune system, initiating a targeted reaction against the pathogens. Naturally, such a sophisticated activation involving cell differentiation/formation takes a fairly long time, typically several days following a stimulus, 4–7 according to a review [91]. In addition, components of the complement system are also included, e.g., mediating the uptake of pathogens by phagocytosis, triggering a number of inflammatory reactions, having a hampering effect on pathogens [91].

### 3.2. Acute Phase Response and Cytokines

Bacterial, and to a lesser extent viral infections, lead to a strong acute phase response (APR). This includes the secretion of especially inflammatory cytokines, but also of APPs and larger proteins stored in blood cells termed pentraxins [92]. Among the main cytokines, though there is some debate about their terminology, there exist chemokines, interleukins, interferons, lymphokines, adipokines and tumor necrosis factors [93]. A main class released by pathogen infected cells are interferons (IFs), especially IFγ from mononuclear inflammatory cells, though others such as TNF-α and IL-1β also presumably play predominant roles and can be secreted from a various tissue cells, especially following stimulation by endotoxins such as LPS [94]. Though they may be produced by a large variety of cells, especially the production by macrophages, B-and T-lymphocytes, neutrophils and monocytes, NK-cells, endothelial and epithelial as well as dendritic and mast cells is important, as they may trigger the production of additional pro-inflammatory stimuli [95].

At least 15 cytokines have been reported to be involved in APR, following their secretion by activated leukocytes and other cells [96], with various functions. These primary agents of stimulated cells are considered the first line of defence of stimulated cells and thus inflammation, and they can be formed within hours of a stimuli. A number of cytokines can act as growth factors (negative or positive), including IL-2, -3,-4, -7, -10, -11, -12 and granulocyte-macrophage colony stimulating factor [97]. Pro-inflammatory cytokines encompass TNF-α/β, IL-1α/β, IL-6, IFN-α/γ and IL-8 [98]. Cytokines with anti-inflammatory nature include IL-1 receptor antagonists, soluble IL-1 receptors, TNF-α binding protein, adiponectin, IL-10 and IL-1 binding protein [99]. Among the major roles of pro-inflammatory cytokines are activating white blood cell precursors in the bone marrow and stimulating the growth of inflammatory tissue fibroblasts and macrophages. Additional functions include angiogenic, growth-factor and osteoclast-activating properties.

At least 33 ILs are known [93]. The most common pro-inflammatory markers include TNF-α, IFγ, IL-6 and IL-8. A more complete overview on cytokines can be found elsewhere [93]. TNF-α, IL-1β and IFγ are needed for inducing the secretion of other pro-inflammatory cytokines (IL-6 and IL-8), as well as fostering prostaglandin, leukotriene and NO production. Following sequestration, IL-6 can bind to other cells via type I cytokine receptor [100], activating various transcription factors such as the JAK/STAT pathway, enhancing pro-inflammatory responses, e.g., via increased APP production. In fact, IL-6 is considered a major agent for the secretion of most APPs by the liver [96]. It also plays a role in the differentiation of CD4^+^ cells [100]. On the other hand, it suppresses mononuclear phagocytic production of IL-1 and TNF-α, thus also exhibiting inflammation resolving properties [101]. IL-6 can further act as an anti-inflammatory myokine [102]. IL-8 is a major chemoattractant of neutrophils and immune cells, activating them. It is also pro-angiogenic and is overexpressed in a variety of cancers [103]. Other cytokines, especially IL-1 and TNF-α, can enhance its expression in many cell types. IL-8 binds to the G protein-coupled receptors cysteine-X-cysteine chemokine receptor 1 or 2, potentially activating several signalling pathways and transcription factors, such as mitogen-activated protein kinase (MAPK) and JAK/STAT, emphasizing pro-inflammatory signals [104]. TNF-α is secreted mostly by macrophages, regulating immune cell activity. Its main role is to mediate acute inflammation, stimulating cells of the endothelium to produce selectins and leucocytes. It also stimulates the liver to produce APPs and acts on muscle and fat-cells [105,106]. Its name originates from its recognized activity to cause necrosis in tumors, but it is also involved in the response to bacterial and viral infections. It can bind to the TNF-receptor 2 (TNFR2), triggering NF-κB translocation to the nucleus, thus being related to inflammation responses, apoptosis, cell differentiation, and production of additional cytokines [107]. It also can trigger MAPK and activator protein 1 (AP-1) activation, important for cytokine transcription. It can further bind TNFR1, activating STAT3 [106], often overexpressed in cancer. Finally, IFγ is best recognized for its activity against invading organisms as it triggers macrophages and MHC II (Class II major histocompatibility complex) expression, important for presenting peptides of foreign origin to T-cells [108], triggering adaptive immune responses. It is mostly secreted by NK-and T-cells [109], can bind to the IFγ receptor and is a major activator of the JAK-STAT pathway, often up-regulated in cancer. Finally, for down-regulating the hepatic APR and removal of circulating cytokines, IL-10 release by Kupffer cells of the liver does play a role, suppressing local IL-6 production [110]. Another compound released in response to bacterial/viral infection is neopterin and its precursor 7,8-dehydroneopterin [111]. It has been shown to correlate with APPs [112], but may represent rather a marker of an active immune system, also in response to cancer [113]. This purine nucleotide derivative is produced by macrophages especially upon stimulation by IFγ.

In addition to bacterial and viral infections, additional triggers of inflammation include xenobiotic/pollutant exposure, radiation, smoke, among others. As cytokines are too large and too polar to pass the cell-membrane, they rather relay signals via cellular receptors to the interior of cells. Of note, cytokines have a very short half-life in blood, i.e., several minutes and thus may not be useful for diagnostic purposes, in contrast to APPs with a half-life of several hours [114], though they would be expected to appear following the cytokines. For example, in patients with sepsis, TNFα typically peaked within 2 days, IL-6 within 4–7 days, SAA within 3–4 days, CRP within 4–6 days, though with considerable inter-individual differences [115].

### 3.3. Acute Phase Response and Acute Phase Proteins

APR refers to the non-specific and complex innate reaction occurring within a few hours after acute infection, tissue injury or trauma. Its purpose is to restore homeostasis and to prevent microbial growth [116]. At the site of injury, pro-inflammatory cytokines are released from tissue. Pro-inflammatory cytokines, especially IL-6, alters APP production in the liver. IL-6 levels increase dramatically during inflammation, stimulating B-lymphocyte proliferation and increasing neutrophil activity [117]. The released APPs may exhibit pro- or anti-inflammatory properties [118]. Examples for pro-inflammatory APPs are CRP, SAA, haptoglobin, and mannose binding lectin (MBL). Examples for anti-inflammatory APPs include albumin, transthyrethrin (TTR/prealbumin) and transferrin.

CRP functions by binding to a receptor (lysophosphatidylcholine) on inflamed dying or dead cells, activating the complement system. Opsonization, phagocytosis, monocyte recruitment, vascular cell activation and thrombosis are some of the consequences [119]. SAA, typically bound to apoproteins in HDL, functions via activating chemotaxis of inflammatory cells, inducing inflammatory cytokine secretion, provoking lymphocyte proliferation [120]. Pro-inflammatory cytokines stimulate SAA production in the liver, where it reacts with HDL to produce a dysfunctional form of HDL, which is then eliminated by phagocytosis. As a consequence, there is a decreased antioxidant and anti-inflammatory activity [121]. Haptoglobin limits haemoglobin (Hb) iron availability for bacterial growth and prevents free Hb from entering in the kidney, as inflammation induces lyses of red blood cells, liberating Hb. This Hb forms a complex with haptoglobin released by the liver. This heterodimer is recognized by CD163 molecules, which results in a dissociation of the Hb-haptoglobin complex. The released iron is reused for Hb synthesis and biliverdin is excreted via urine as bilirubin [122]. Mannose-binding lectin (MBL) circulates in the blood, recognizing pathogen-associated molecular patterns (PAMPs), i.e., specific molecules from pathogenic microorganisms. The route that MBL describes in the complement system is the so-called lectin route, which exists as a third activation possibility in addition to the classic (antibody mediated) and the alternative pathway (complement C3b mediated) [123]. Lectin binds mainly to mannose or N-acetylglucosamine, activating specific serine proteases. These in turn cleave the complement factors C2 and C4, whose subgroups then cleave C3. From this point on, the cascade follows the classic route to “perforate” bacteria [124].

Inflammation decreases the concentration of circulating albumin, likely by increased capillary permeability, increased interstitial space and volume, also effecting the circulatory half-life [125]. Indeed, albumin is the main extracellular antioxidant, considered to be responsible for 3/4 of the extracellular antioxidant potential [126]. In serum, it represents more than 50% of plasma proteins [127]. Somewhat similar considerations are true for prealbumin, also termed transthyretin. This extracellular protein aids in the transport of vitamin A in form of retinol binding protein (RBP) and thyroxin in the plasma and its role in OS has been highlighted [128].

Another marker discussed for inflammation is transferrin, responsible predominantly for iron transport in serum. Its concentration decreases during inflammation, causing iron to be stored as hemosiderin in the liver. This feedback has possibly been developed during evolution to protect against micro-organisms, as a low iron status may limit the spread of several bacterial and parasitic infections [129]. However, transferrin is predominantly associated with anemia/malnutrition [130]. Finally, an also frequently investigated APR is procalcitonin, precursor of the calcium regulating hormone calcitonin. It is produced in the thyroid glands and can be released by a variety of parenchymal cells in different organs, especially from the lung and the intestine upon bacterial infections [131]. Its role in sepsis as a marker with noxious properties has been highlighted [132].

### 3.4. Role of Transcription Factors, Especially NF-κB, in Inflammation

NF-κB is a small family of inducible transcription factors, present mostly in its inactive form in the cytosol of mammalian cells. This factor is important for regulating immune responses, cell proliferation, apoptosis and inflammation [133]. Most notably, NF-κB can foster the expression of a large number of target genes, including the pro-inflammatory cytokines IL-6, TNF-α, IL-1β, in addition to a number of complement proteins and APPs (Figure 2). NF-κB is especially found in B-lymphocytes. The family consists of five different subunits that bind to a core of 300 amino acids [134]. Two of the proteins together can form a homo-or hetero-dimer. Five subunits are currently known: NF-κB1 (p50 or p105), NF-κB2 (p52 or p100), RelA (p65), RelB and c-Rel.

NF-κB can be regarded as a master switch of inflammation. When cells are subjected to extracellular stimuli (e.g., ROS), pro-inflammatory cytokines (TNF-α, IL-1, bacterial LPS etc.) [135] are released and this pathway is activated following their binding to cell-surface specific receptors. NF-κB can also be down-regulated, with consequential suppression of pro-inflammatory pathways. Such appears to be the case for NF-κB downregulation by cortisol, responsible for its immuno-suppressive effects [136], and also by certain phytochemicals such as phenolics [137], though in general these processes are less well understood than its activation [138]. For NF-κB activation in the canonical pathway, in a first step, the inducer binds to a cell membrane receptor that will be activated and subsequently will phosphorylate the inhibitor of NF-κB (IKB-α). In physiological state, IKB-α binds to NF-κB, rendering the dimer of p50 and RelA inactive. After phosphorylation, the NF-κB/IKB-α complex dissociates. The phosphorylated IKB-α is ubiquinated and degraded by the proteasome. NF-κB is translocated into the nucleus, binding to the κB motif on the DNA, triggering the transcription of genes that include adhesion proteins and cytokines [139]. NF-κB has been shown to be involved in activating numerous downstream components, from pro-inflammatory cytokines (e.g., IL-6, 8, TNF-α), chemokines (e.g., MCP-1), adhesion molecules (ICAM-1, VCAM-1), anti-apoptotic factors, cell-cycle regulators, among other. As some of these mediators such as TNF-α and IL-1 are further inducing NF-κB translocation, this self-amplifying cycle produces even more cytokines, attracting more neutrophils, fostering a stronger pro-inflammatory reaction [140].

Due to its central role in inflammation and the immune system, NF-κB activation is implicated in various diseases. In autoimmune diseases such as rheumatoid arthritis (RA) or IBDs, and especially in cancer, NF-κB activates signal transducer and activator of transcription 3 (STAT3), important for many oncogenic functions [141]. NF-κB is also involved in the differentiation of various T-cells from naïve CD4^+^ cells. For instance, more naïve CD4^+^ T-cells can be differentiated into TH17 cells, involved in triggering tissue destruction in autoimmune diseases, such as in RA [142]. TH17 also triggers more neutrophils to enter the blood, fostering pro-inflammatory processes. In principle, there are two ways to interrupt this self-amplifying cycle. The first one is avoiding triggers of this process, including ROS, pollutants and infections. The second one is subduing inflammation, e.g., by administering dietary suppressants including fish oil, vitamin E, secondary bioactive plant compounds, or by drugs such as cortisol.

In addition to NF-κB, STAT proteins such as STAT1 are involved in immune responses, including iNOs, COX, VCAM-1 and ICAM-1 expression [143]. It is activated by various cytokines such as IL-6 and especially IFα and IFγ. They are part of the JAK/STAT pathway, in which janus kinase (JAK) is released from cellular receptors, phosphorylating STAT, resulting in its dimerization—or dimerization as heterodimers with e.g., STAT3 and translocation to the nucleus, binding to GAS termed parts of the DNA.

## 4. Markers of OS, Relation to Disease and Practical Aspects

### 4.1. European Food Safety Authority (EFSA) Accepted Markers

Numerous studies have revealed chronic OS involvement in a variety of chronic diseases, namely cancer, CVD, atherosclerosis, diabetes, arthritis, neurodegenerative conditions, as well as respiratory, kidney, and liver diseases [144,145,146,147]. The risk for these pathological conditions increases with age and OS is thought to be a key factor in the aging process and age-related diseases [148,149]. OS markers are therefore valuable for determining and monitoring the biological redox balance, the state and prevalence of disease and studying effects of interventions such as administering antioxidants. Many studies have focused on identifying suitable markers of OS, and numerous have been proposed (Table 1). These have included measuring directly ROS species, measuring their induced damage such as by MDA or oxidized DNA/RNA or proteins, or measuring more indirectly enzymes involved in the antioxidant-oxidant balance such as SOD, CAT or GPx, or antioxidant compounds such as GSH or TAC of a serum specimen [55].

Interestingly, a scientific opinion has been published by EFSA regarding markers sufficient for the substantiation of EFSA health claims in Europe [150], which are typically only granted upon the provision of very solid scientific evidence. To claim efficacy of a dietary component against OS in human studies, it was recommended to include at least one of the EFSA recognized markers, plus additional markers. Antioxidant markers such as total antioxidant capacity, FRAP, ABTS etc. and enzymes such as SOD, CAT, GPx were not among the recommended markers, due to lack of data, representing rather antioxidant than oxidant effects, or due to technical limitations. The EFSA accepted markers include:
(a)Oxidative damage to proteins assessed by direct measures, such as LC-MS-MS, to detect e.g., protein tyrosine nitration products;(b)For oxidative damage to lipids, F2-isoprostanes, also measured by LC-MS-MS (not by ELISA due to cross-reactivity); also oxLDL by immunological methods; lipid hydroperoxides by chemiluminescence; but not e.g., MDA/TBARS (though seen as a supportive measure, i.e., together with an accepted marker), LDL oxidation ex vivo;(c)DNA damage, as assessed by the COMET assay; not 8-OH-dHG (though accepted as a supportive marker).


### 4.2. Direct Markers of ROS—Primary Radicals, Hydroperoxides

Measuring originating ROS species would be the most direct and possibly accurate way to determine ROS, as these are the primary species involved in causing oxidative damage such as oxidation of macromolecules. However, this is severely impeded by their short-lived nature, and their instability during sample preparation and/or storage. Some methods exist, such as measuring O_2_^−^ by EPR-spin trapping, termed the “gold standard” [16]. EPR detects molecules with unpaired electrons and can be used ex vivo such as in isolated cells as in whole blood samples, or certain subcellular fractions including mitochondria. The method is analogous to NMR, though electron spins are exited, not atomic nuclei. This method requires an EPR spectrometer, working in the microwave-wavelength range [202]. The spin-trapping of the free radical happens with a separately added molecule, i.e., the double bond of the “spin trap”, to form a compound with a long enough half-life to be detected. However, due to the difficulties regarding the need to measure fresh samples and the marginal availability of EPR instruments, this method has not found many clinical or even research applications. It has been proposed and used in some neurological disease such as to study human prion diseases [151]. In an interesting but small-scale study with 100 middle-aged subjects, capillary blood was measured by EPR, following the addition of a spin probe. With increasing age, ROS production increased, and EPR measures correlated well with PCs and TBARS [152].

As a compromise and a more practical approach, more long-lived species can be assessed. These are especially lipid hydroperoxides such as phosphatidyl-choline hydroperoxides (PCOOHs) and cholesterol hydroperoxides. For EFSA, these constitute appropriate markers, though their measurement together with F2-isoprostanes is preferred [150]. Cholesterol hydroperoxides and oxysterols are more recent markers, detectable e.g., by HPLC, in combination with chemiluminescence or UV, respectively [203]. Especially 7β-hydroperoxycholesterol has been mentioned as a marker of OS [204], but it has not found its way into clinical trials. All of these products are expected to be formed rather rapidly following OS exposure, based on in vitro simulated experiments, but no in vivo data appears available.

PCOOHs would be situated at the outer layer of lipoproteins, they can be measured from blood samples following a lipid extraction. However, laboratory analysis requires sophisticated instrumentation (such as LC-MS), and standards may suffer from instability. More recently, phospholipid oxidation products are determined by antibodies including ELISA kits, which have been employed to detect oxidation of cholesterol within lipoproteins, i.e., oxLDL (Section 4.3). Nevertheless, EFSA has recommended this compound as a valid marker of OS [150]. Another limitation is that little data exists regarding their storage stability. It was emphasized that storage at 4 and even −20 °C resulted in increased formation of PCOOHs, while shipment at −78 °C did not increase levels significantly [205]. However, processing samples under oxygen exclusion was recommended. The formation and reactions to PCOOHs to secondary products such as 4-hydroxynonenal (4-HNE) are beyond the scope of this review and have been reviewed elsewhere [36], but lipoxygenases and hemoproteins have been reported to play a role [206].

Regarding clinical trials, no meta-analysis exists to our knowledge, and few human studies have reported measuring PCOOHs. However, their role as biomarkers in human diseases has been emphasized [37]. It has been recognized that increased PCOOH concentrations can be found with ageing [168] and in dialysis patients with diabetic nephropathy [169]. Their concentrations have also been associated with poor glycemic control in diabetic individuals [207]. In a mice study, higher concentrations of PCOOH were found in the liver (but not the brain) following chronic alcohol consumption, which was ameliorated by simultaneous dosing with curcumin [208]. The strong negative effect on organs have been demonstrated in mice receiving oxidized phosphatidylcholines in their diet for 2–4 weeks. Compared to those being fed unoxidized phosphatidylcholines, higher levels of hemolysis were detected, with concomitant increased MDA in the heart and other organs [209].

### 4.3. Oxidized Lipids, Proteins, Lipoproteins

#### 4.3.1. TBARS and MDA

MDA appears to be the major abundant product of LPO, though it is not the most reactive product—for instance HNE or acrolein have been reported as being much more reactive, though HNE may be 80 times less abundant [210]. Of note, MDA can also be present in food items (those rich in lipids) [211] and also absorbed and thus this marker alone may lack specificity. Consequently, MDA and other aldehydes are often used as secondary markers for LPO. Their detection is often achieved by photometrically determining them as thiobarbituric acid-reactive substances (TBARS). MDA can be more selectively detected by chromatographic separation of the TBARS, e.g., by HPLC-fluorescence detection [212]. The TBARS method has been frequently applied to especially detecting MDA in serum/plasma and urine.

4-HNE, another reactive, unsaturated aldehyde, originates especially from ω-6 fatty acids [213], forms protein adducts and induces apoptosis [214,215]. It is thus of biological importance, however, due to its reactivity and lower stability, it is less commonly assessed than MDA. Acrolein, formed also during lipid peroxidation from PUFAs, is even less stable, forming adducts with other biomolecules, such as with proteins and DNA [216]. However, it can also origin from the diet and constitutes an environmental pollutant (released e.g., during combustion), thus lacking specificity [216]. Nevertheless, as all three compounds may be of interest, methods to detect all three in urine have been developed [217].

Various studies have shown the relation of MDA to chronic diseases. For example, in recent meta-analyses, elevated MDA levels were associated with Parkinson’s disease vs. control subjects [17], in patients with chronic obstructive pulmonary disease (COPD, [158]), and treatment with statins significantly reduced MDA [218]. In another meta-analysis, also elevated age-related macular degeneration, related to low carotenoid concentrations in the macula, was associated with higher MDA in plasma/serum, though heterogeneity between studies was large [157]. As TBARS measures largely MDA, similar results in meta-analyses are expected. In a meta-analysis with T2D subjects receiving vitamin E and C supplements for 8–24 weeks showed a significant reduction of circulating TBARS/MDA [159]. MDA levels may adapt quickly to external stimuli—following injury of the spinal cord in a rat model, MDA extracellular levels increased after 2 h, reaching a maximum at 5 h [219]. Acrolein and 4-HNE have also been related to various diseases, including their increased levels in pre-clinical Alzheimer’s individuals [220] and atherosclerosis [221], though meta-analyses have not been reported.

A disadvantage of the MDA/TBARS test is the limited stability of samples during storage. Levels of MDA may increase ex vivo due to ongoing oxidation if samples are not stored under oxygen and light exclusion and at cold temperature. For instance, MDA exponentially increased when stored at −20 °C for >3 weeks, and by almost 10-fold after 1 year [156]. When storing whole blood samples on ice for 36 h, MDA concentration was ca. two times higher compared to fresh samples. MDA increases were even higher compared to those of F2-isoprostanes [222].

Discrepancies of measurements can also originate, depending whether spectrophotometric, fluorimetric (more sensitive but prone to quenching effects) or chromatographic methods (being typically more specific but time consuming) are used. For instance, when comparing HPLC based MDA derivatization with a spectrophotometric (TBARS) method, 25% higher values for the latter method were found, with a low correlation (R = 0.284) between the two methods [223].

#### 4.3.2. Isoprostanes

The most frequently studied but not necessarily most abundant isoprostane is 15-F_2t_-isoprostane [224]. Due to various nomenclatures, this compound is also termed 8-iso PGF2α, 8-epi PGF2α or just 8-isoprostane. Of note, as also E and D-isoprostanes exist, the total number of isoprostanes is 3 × 4 × 16 or 192 isomers [225]. Their advantage as markers is that they are specific for OS, are fairly stable during storage and can also be detected in free form in urine (in addition to plasma or other tissues), where they are present at about 20–100 times higher concentrations (ca. 1000 ng/L) than in plasma [163]. However, a 24-h collection is advised in order to be more representative than spot-urine samples [225]. A main limitation, which was only realized in recent years is that there is an OS independent pathway for generating F2-isoprostanes, catalysed by prostaglandin endoperoxide synthase [163]. It has been proposed that measuring total or esterified F2-isoprostanes rather than free ones alone would be a superior marker of OS, as esterified arachidonic acid is not reacting with prostaglandin endoperoxide to F2-isoprostanes.

Problems may also arise due to the large number of isoprostanes, with cross-sensitive ELISA-kits. Though refuted by many ELISA producers, it is still an ongoing debate [225]. Though correlation, according to some studies, between ELISA and MS is good, ELISA may generate >50% higher results [162]. As highlighted by EFSA, MS-based (GC-MS or LC-MS) methods for unique detection are preferred [150], though for practical reasons, a large number of ELISA-based studies exist, which do not necessarily allow a comparison between them due to variable cross-reactions. Sample storability poses another problem, as F2-isoprostanes may be produced ex vivo. When leaving blood samples for 36 h on ice, F2-isoprostane levels doubled [222]. Stability of plasma samples for half a year at −80 °C was shown, though one study resulted in increased [160] and one in decreased levels [161] following 2-3 cycles of freezing/thawing.

Many clinical studies have supported the role of F2-isoprostanes as a valid biomarker of OS. It also appears to be formed rapidly. In a study with rats receiving (i.p.) carbon tetrachloride, peak concentrations of F2 isoprostanes, similar as for MDA, appeared already after 2 h, and depended on the concentration of CCl_4_ [226]. Regarding human chronic diseases, a meta-analysis of several diseases was done by van t’Herve et al. [163]. While some conditions such as hypertension, smoking and the MetS were associated with rather small increases of F2-isoprostanes, strong increases were seen for kidney pathologies and respiratory disorders. Other meta-analyses found relations between cardiac arrest and F2-isoprostanes [164], positive airway pressure treatment in subjects with sleep apnoea [165] and depression [166]. One meta-analysis including 350 studies investigated means to lower 8-iso-prostaglandin F2α [227]. While supplementing antioxidants showed only small benefits, larger changes were found with interventions of antibiotics, lipid-lowering drugs, and life-style changes.

#### 4.3.3. OxLDL

OxLDL, following oxidation of phospholipids and cholesterol within LDL particles, as well as of their apo-protein amino acids, has become a frequently used marker of OS, especially due to its relation with atherogenic processes, contributing to foam cell formation [228]. A first challenge with this method is standardizing LDL isolation and determining the oxidized moieties, which may be rather variable. In addition, the formation of LDL particles and oxLDL and their clearance may vary largely, even within subjects [229]. It is also apparent that changes in LDL can include many modifications, from minimally to extensively modified LDL, from lipid oxidation products such as prostaglandins and aldehydes to cholesterol oxidation to modified amino acids [230,231]. A special case that has received attention is MDA-modified LDL, in which MDA reacts with lysine residues in ApoB, which can be recognized by a specific antibody (4E6, [231]). To complicate the picture, some oxidized lipids and sphingosines may even have anti-inflammatory properties.

There is no single accepted method to determine oxLDL, but immunological ones based on monoclonal antibodies are the most common, also accepted by the EFSA [150]. Another option is measuring lipid reaction products by e.g., TBARS. Regarding antibodies, two main options have been reported in the literature, which rely on detecting specific modifications of amino acid residues in LDL particles, namely aldehyde-modified lysine of apoB (aka MDA-modified LDL) or oxidized phospholipid-residues (e.g., DLH3 against oxidized phosphatidylcholine) [232]. Due to the contained peroxides, oxLDL particles are far from stable, which is a limitation for sample storage and using oxLDL as a marker. However, using the widely employed 4E6 antibody, whole blood samples were stable for at least 36 h on ice [170]. Storing plasma may pose a challenge, though when using anti-freeze reagents, stability of several months at −20 °C was achieved [171]. Indeed, many kits detail a storability of plasma samples free of erythrocytes and buffy coat layer of 3 months when kept frozen. Related to the stability of LDL are also the ex vivo conducted copper-induced oxidation experiments, which can be measured over a time course of 2–3 h [22]. The lag-time until rapid peroxidation is measured spectrophotometrically. Though this is a somewhat non-physiological assessment, it appears to give an indication of the antioxidants present in LDL particles. However, the measurement of samples is time consuming and requires previous LDL isolation. However, positive effects of dietary interventions can be measured, such as following supplementation with soy-fortified tomato juice for 8 weeks [233].

Regarding clinical studies, few systematic reviews or meta-analyses exist. Gao et al. reported a meta-analysis of 12 observational studies, targeting atherosclerotic CVD [23], showing that higher oxLDL was significantly associated with disease outcome. In a systematic review of all types of studies (*n* = 18) relating oxLDL and CAD, a positive association between anti-oxLDL antibodies and CAD was found [234]. OxLDL was elevated in a meta-analysis of 6 studies targeting sleep apnea [172]. In a network meta-analysis [173] including 4 RCT studies on the effect of olive oil on oxLDL, higher intake was associated with decreased oxLDL.

#### 4.3.4. Advanced Oxidation Protein Products (AOPP) and Protein Carbonyls (PCs)

AOPPs are oxidation products of proteins, e.g., of plasma albumin, following their oxidation by chlorinated oxidants (HOCl, chloramines). Major groups include PCs, in addition to dityrosine and pentosidine [235]. AOPPS are typically measured by colorimetric assays (often containing acetic acid and potassium iodide), based on measuring products with chloramines at 340 nm [236]. PCs can be detected by immunological such as ELISA methods [43], spectrophotometric methods and MS [179], including proteomics [237]. ELISA kits, offered by several companies, often include a derivatization step with dinitrophenlyhydrazine (DNPH), followed by staining with an anti-DNP antibody. Similarly, spectrophotometric methods also require derivatization, and DNPH adducts can be detected at 360 nm. Other derivatizations exist but are out of the scope of this review, thus the reader is referred to additional literature [179]. AOPP stability for 6 months at both −20 and −80 °C has been shown [174]. Regarding PC stability, isolated protein fractions were reported to be stable for 3 months at −80 °C [238]. In general, PCs are assumed to be rather stable. Commercial kits judge the stability of plasma samples to at least 1 month at −80 °C [239], which has also been stated by others [178], though data is scant.

AOPPs have been associated with a variety of diseases, though no meta-analysis is present. For instance, elevated levels were found in individuals with acute coronary syndrome [175], Crohn’s disease [176], diabetes [177], among other. Likewise, PCs have been related to e.g., obesity, neurodegenerative diseases, lung and kidney diseases, and diabetes [240,241,242]. A meta-analysis studying the relation of bipolar disorders and PCs in >970 patients failed to find a significant relation [243]. Mild OS and increased PCs were detected in an RCT after exercise [244]. In early Alzheimer’s disease, elevated PCs were detected in patients compared to age-matched controls [180]. In an RCT with T2D individuals, administering carnosine as an antioxidant significantly reduced PCs vs. placebo [181]. PCs have also been shown to increase within 5–7 days post-operation, thus AOPPs and PCs may be observed after some time-lag of the original stimulus [245]. Thus, though there is less data compared to other OS markers, PCs constitute an interesting choice with clinical relevance, though the diversity of analytical methods impedes comparisons across studies.

In summary, both oxLDL and F2-isoprostanes, the latter when detecting total or esterified forms, in conjunction with unequivocal detection by MS-based techniques are EFSA recommended and useful methods to be employed in the assessment of OS, with possibly longer stability on the F2-isoprostane side, though with the less instrument-intensive antibody measurements to the advantage of the oxLDL. For AOPPs and PCs, too little clinical data is available, while MDA may be a meaningful additional measure of OS, despite its possible origin also from the diet.

### 4.4. Oxidized RNA/DNA

#### 4.4.1. 8-OH-dG

This compound is among the most frequently used biomarkers related to oxidative nucleic acid damage and its presence can be measured in a large number of specimens, including lymphocytes, leukocytes, the brain, plasma and urine [246], e.g., via immuno-assays (ELISA), or LC [247]. Measurements in urine offer a more simple matrix, with up to 100–1000 times higher concentrations compared to serum/plasma [248]. A main advantage rests in the fairly long stability, which may extend to years of storage [19], even at non-optimal low temperatures (above −80 °C). In this last study, samples were stable for at least 2 years at −80 °C. Another advantage is that it can be measured both in urine and blood, i.e., in free form, though it may also be analysed in cells after lysis, assessing total RNA/DNA breakdown products. The time course of in vivo is not clear, but in in vitro trials, rapid formation of products, e.g., chromium induced, has been shown (within 20 min) [249]. Another advantage is that 8-OH-dG is biologically active, as it can pair with cytosine and adenine during DNA replication, resulting in mutations [250]. Thus, this molecule is not only a passive marker of ROS. While chromatographic measures are very specific, most antibody-based methods may show cross-sensitivity. In some cases (e.g., “Cayman RNA/DNA oxidative damage kit”), this is desired, offering a broader overview on degradation, originating from RNA and DNA.

8-OH-dG has been correlated to several diseases. For instance, solid tumor survival prognosis was inversely correlated in a meta-analysis with >2000 patients with higher 8-OH-dG [153]. High levels were associated with a 2-fold higher mortality risk. In another meta-analysis with case-control and prospective studies of CVD [18], higher levels were found in CVD patients (*n* > 800) vs. controls (>1100), both based on urine and plasma. Higher urinary levels were also detected in a meta-analysis in smokers [251]. However, another meta-analysis failed to find significant correlations between Alzheimer’s disease and 8-OH-dG levels in tissue [252]. Recently, an interesting relation to vitamin D levels in obese subjects was also shown [253], with higher levels being correlated to lower vitamin D but higher visceral fat.

#### 4.4.2. COMET Assay and γH2AX

Another popular marker of DNA breakdown is the COMET assay, or rather its alkaline version, which is more sensitive compared to the neutral one, as it allows studying single strand, not only double-strand breaks, and is also recommended by EFSA [150] (as opposed to 8-OH-dG), at least when using a modified version. The typical test is carried out by electrophoresis, following lysis of cells with specified buffers removing cell membranes, cytosol, and nucleoplasm. The DNA fragments travel to the anode in a cast gel, leaving a comet-like structure. The size and shape of this tail from individual cells when compared to control cells under e.g., fluorescence microscopy allows estimating the amount of DNA damage. The modification also proposed by EFSA includes measuring oxidized bases, i.e., pyrimidines following an endonuclease III treatment, or detecting damaged purine bases with e.g., formaminopyrimidine DNA glycosylase [254,255]. The COMET assay has been known for some time, being developed in 1984 [256], detecting (in the alkaline version) DNA double and single strand breaks and sites sensitive to alkaline solutions from several 100 to several 1000 breaks per cell [257]. This method has been applied to many types of cells, and its sensitivity to detect DNA damage is regarded superior when compared e.g., to the TUNEL and HALO assays. A limitation is that small deletions may go unnoticed and that results across studies are hard to compare, as time and voltage of electrophoresis, buffer conditions and temperature and the type and concentration of the gel can influence results [258]. It can further be criticized that the assay rather measures potential, not actual damage, as the extent of damage may still be kept in check by DNA repair mechanisms, another reason to employ the modified alternative. Also, cells are needed such as leukocytes, and sample preparation and storage conditions have shown to influence results. Fresh samples may pose less risk for variability of results than frozen ones [259]. Nevertheless, slow freezing procedures in DMSO and storage at −80 °C have become more common and accepted [254].

The Comet assay has been employed in a variety of clinical studies. For example, it was shown that diabetic subjects had >2 times more strand breaks compared to controls. However, a large variability, also within-persons, was highlighted [260]. Nevertheless, compared to other markers such as 8-OH-dG, fewer meta-analyses are available, possibly due to the difficulty to compare non-absolute results across studies, and due to the more complex analytical investigation. An early meta-analysis of 38 studies showed that smokers had a significantly higher level of DNA damage than non-smokers [154]. A systematic review exists on the effect of dust, asbestos and other fibers on DNA damage. Due to the large heterogeneity between studies, no meta-analysis was done, and effects appeared non-significant for the Comet-assay [261]. The lack of studies employing this technique was also criticized. Another meta-analysis with results from 35 studies using the Comet assay highlighted short-term (2 h–1 day) increased damage post-exercise, though long-term results did not indicate increased damage [155]. Of note, another meta-analysis compared the accuracy of the Comet, Tunel and sperm chromatin dispersion (SCD) test to detect sperm DNA damage as a marker of infertility [262], with the Comet assay providing lowest accuracy.

Another, more novel marker of DNA breakdown is γH2AX, a phosphorylated histone variant (being in close proximity to DNA) formed following double strand breaks, after phosphorylation of a serine residue. This reaction is a rapid one, within a few minutes after exposure to e.g., ionizing radiation [263]. Normally, the amount of strand breaks is proportional to the amount of γH2AX formed, which can be measured by antibodies. So far, it has mostly been used as a biomarker for cancer [182,183]. It was argued that it is rather a marker of apoptosis and not of OS [184]. Further studies targeting additional diseases are needed to investigate γH2AX.

Taken together, though DNA damage by the Comet assay, especially when measuring not only strand breaks but also damage to the DNA bases being an EFSA accepted marker, the lack of absolute values and the variability due to analysis conditions have precluded this marker from being utilized in clinical practice. On the other hand, 8-OH-dG is quite stable and has been employed in many clinical studies, but results, due to cross-sensitivity, may differ between methods. Nevertheless, it is a promising marker, awaiting further validation in future studies. A further new development may be to assess also the influence of genes involved in DNA repair, concomitantly with 8-OH-dG [264].

### 4.5. Antioxidant Enzymes and Antioxidants

#### 4.5.1. Antioxidant Enzymes

SOD in humans exists in three isoforms, present in the cytoplasm (SOD1, soluble form, requiring Cu and Zn as cofactors), in the mitochondria (SOD2, requiring Mn), and in the extracellular space (SOD3, requiring Cu or Zn). As the latter is more easily accessible, it is this isoform which is most often measured. CAT, an iron dependent enzyme, is usually located intracellular in peroxisomes, and is often measured in red blood cells. GPx exists also in various isoforms. GPx1 is present in peroxisomes, mitochondria, and the cytosol, GPx2 is found in the intestine, in the crypt epithelium, and GPx3 is found in plasma [265]. Also GPx4 exists, another membrane-bound enzyme often associated with phospholipids and thus with an affinity for cholesterol hydroperoxides bound to cell membranes [266]. Additional isoforms (GPx5–8) have been reported but appear to play less important roles. At least the GPx1–4 isoforms are selenium dependent. As the GPx3 is easily accessible, this one has most frequently been measured in studies [265]. Peroxiredoxins exist in 6 isoforms in humans and are cysteine dependent peroxidases, present in mitochondria, the cytosol, the ER and the plasma membrane [267].

SOD, CAT and GPx seem fairly stable, and analysis is typically carried out spectrophotometrically. No significant differences when blood samples were stored on ice between directly measured activities in erythrocytes and those measured after 48 h were found [268]. Also, no differences were realized for red blood cell hemolysates between measurements from fresh samples or after 21 months of storage at −80 °C for SOD and GPx (and GR) [185]. Again, activity may differ between person, varying age, gender, and smoking status [269], with SOD being lower with age and CAT being lower in smokers. The authors also proposed references values in blood for healthy males and females.

Regarding disease association, in a meta-analysis on gastric cancer, a significantly lower SOD activity in plasma and erythrocytes was found compared to healthy individuals [270]. In another meta-analysis, mitochondrial SOD2 was lower in subjects with schizophrenia vs. controls [271]. In a meta-analysis of observational studies, lower levels of SOD, CAT and GPx were found in persons with coronary heart disease (CHD) [186]. Decreased activity of these enzymes was also observed in a meta-analysis in subjects with Parkinson’s disease [187]. For peroxiredoxins, little data is available, but elevated levels of several isoforms were measured in plasma of individuals with T2D [267], and prevention of apoptosis of pancreatic-beta cells was reported [272].

#### 4.5.2. Antioxidant Tests

While GSH is primarily found intracellular and therefore has been mostly disregarded by antioxidant measures of plasma/serum, most other constituents can be aimed to be assessed in a combined assay as TAC. A vast amount of literature exists on markers attempting to measure TAC in various biological samples/extracts, including serum/plasma [273,274] and urine [275].

For measuring TAC in plasma, a multitude of tests exists, mostly photometric methods that are available to most laboratories, allowing also a high sample throughput. Most employ aqueous solutions, but some tests allow measuring TAC in a lipid fraction/extract of plasma/serum, such as the ABTS test. Both measurements should be carried out in order to obtain a more complete picture. Neglecting the lipid fraction would overlook many components interacting with lipid peroxides, such as tocopherols and carotenoids. Most tests function by quenching a radical producing agent by the sample, which would otherwise react with a reducing agent. A full presentation of methods is beyond the scope of this review, but prominent methods include FRAP, ABTS, ORAC, TEAC, DPPH, among other. Most tests are based on electron transfer (e.g., ORAC), hydrogen atom transfer (e.g., FRAP) and metal (e.g., Fe^2+^) chelation. For further literature the reader is referred to more comprehensive reviews [276,277,278]. However, the large number of tests existing and the various laboratory testing conditions are a major challenge [190]. There is no accepted reference method. Instead, it is usually advised to measure antioxidant capacity by at least 2 complementary techniques, e.g., one based on hydrogen transfer and one on metal chelation, due to the numerous molecules involved in the quenching reactions. TAC has been shown to respond to mid-term interventions, such as to dietary intervention. For instance, a 14 day intervention with a high antioxidant diet improved TAC in plasma in triathletes (by about 25%), along with SOD [279]. However, as many other exogenous compounds do also possess antioxidant activity (uric acid, albumin), the effects achievable are considered small, if any.

From a clinical perspective, reduced TAC has been observed in a variety of diseases, though most studies were of small scale. In a meta-analysis examining all-cause mortality, a significant lower relative risk was found for individuals with higher FRAP and TRAP status, though the analysis was only based on three and two observational studies, respectively [188]. In hemodialysis patients, significant higher levels of TAC were found compared to healthy controls based on FRAP and 2 test kits, however TAC decreased following hemodialysis [189]. Diabetic vs. healthy subjects (*n* = 90 participants) showed lower levels of TAC (determined by FRAP) [190]. A general criticism of measuring TAC and drawing conclusions on redox-status is that it depends on many factors, endogenous and exogenous [278], and that homeostasis is rather strictly controlled, so that it may not be linked to specific diseases. In addition, intracellular measurements are typically neglected. As with antioxidant enzymes, only an indirect insight into OS is obtained, and the large number of different testing methods greatly impedes the setting of any reference values.

### 4.6. Transcription Factors—Nrf2

Several kits (mostly ELISA) for detecting Nrf2 exist on the market, and 10–20 pg/mL in a nuclear extract of 1–2 µg protein per (96-)well can generally be detected; PCR may also be used, but due to limited mRNA stability, measuring ARE sequences via reporter assays including luciferase may be superior [24]. The advantage of detecting Nrf2 is that it is involved in many antioxidant activity processes, thus its measurement may yield a more integrative, global insight into antioxidant responses. However, a disadvantage is that it is challenging to compare results across trials, and that the measured expression does not directly reflect antioxidant status. For instance, though Nrf2 may be upregulated, this could reflect a strong OS insult and a reaction to it, or merely a high baseline activation state, making interpretation difficult. In general, only a relative quantification against a standard of unknown concentration can be carried out. Also, stimulation may only persist for a short time and not in all tissues, making it difficult to be used in research and in clinical practice. Nrf2 is mostly expressed in brain, ovaries, gastrointestinal tract and bone marrow [280]. In a rat model of OS caused by intracerebral haemorrhage, Nrf2 expression increased 2 h after insult, peaking at 24 h, and returning almost to baseline after 10 days [281]. In relation to this, HO-1 started to increase after 8 h, peaking at 3 days. In whole-cell extracts, both free Nrf2 and the complex with Keap1 may be detected, though only the first one would be potentially translocated to the nucleus. Even if unbound Nrf2 is measured in the cytosol, not all free Nrf2 may travel to the nucleus, and isolation of the nuclear fraction is thus recommended. Some antibodies detect only Nrf2 bound to DNA in the nucleus, avoiding the problem of detecting Nrf2 bound to Keap, as this inhibitor generally stays in the cytosol [282]. The stability of Nrf2 is likely limited, and sample storage at −80 °C advised, though no data on long-term storage stability is available. ELISA standards appear to be stable for 1 year at −20 °C [191].

Regarding diseases, Nrf2 was investigated in a meta-analysis of 7 studies for its prognostic value in breast cancer [26]. Patients overexpressing Nrf2 had lower overall survival. Nrf2 was measured both by mRNA expression and by immunological methods, though it was not reported if Nrf2 was measured in the nucleus. Another meta-analysis of 17 studies investigated Nrf2 in patients with solid tumors [192], all measured Nrf2 immunologically. Again, no explanation on cytosolic vs. nuclear detection was provided. Also in this analysis, a higher Nrf2 expression was related to reduced survival. Somewhat surprisingly, in a meta-analysis of transcriptomic data summarizing ARE related gene expression, 31 genes were down-regulated in Alzheimer’s and Parkinson’s disease patients, despite Nrf2 being upregulated [193], possibly related to MAFF overexpression, a transcription factor that may act as transcriptional repressor. Peripheral blood mononuclear cell (PBMC) Nrf2 expression did not differ between persons with normal weight and those with obesity or insulin resistance [283]. However, in Mexicans, certain polymorphisms in Nrf2 showed to correlate with obesity [284]. Some authors have suggested that activators of Nrf2 (e.g., the drug protandim), could be beneficial for CVD [285]. It may be speculated that a strong long-lasting overexpression is rather detrimental, while some upregulation may be beneficial when antioxidant status is low, and/or that other regulatory factors do also play a role. This has also been highlighted in a recent review, concluding that activation of Nrf2 showed some success in clinical models, while many diseases including certain cancers exhibit elevated Nrf2 levels. Also the potential to influence Nrf2 by certain drugs was highlighted, at least dimethyl fumarate is FDA approved for patients with multiple sclerosis [286]. Clearly, more studies are needed to understand the regulation of Nrf2 in disease; its role of a marker is yet premature. As no absolute value is usually obtained, and large inter-individual differences appear to exist, this marker may be of interest when following expression in a certain tissue or blood cell type of an individual over time.

### 4.7. “Composite” Indices, -Omics Based Markers

A recent overview on indices of OS was published by Sanchez-Rodriguez et al. [27]. A few indices have received the majority of attention and are briefly introduced here. These include the OSI (OS index) and similar ones such as the OSS (OS score), thiol ratios and glutathione ratio. The latter measures the balance of GSH vs. its oxidized state (GSSG). Due to its intracellular presence, this marker normally requires assembling red blood cells, leukocytes or other types, though it has been used also in extracellular fluids such as saliva [287]. A challenge is the pre-analytical isolation of the fractions to be measured without changing the GSH:GSSG ratio [198]. Therefore, this technique has not been used much in clinical practice or even research studies. A common assay is the GSH-recycling assay, which reduces oxidized glutathione to its reduced form by glutathione reductase/NADPH. The reduced form then reacts with DTNB to form TNB which can be measured spectrophotometrically [288]. No meta-analysis has been carried out on this marker. Clinical studies employing the GSH:GSSG ratio have been reviewed [27], listing two dozen individual studies, most of them small-scale. A number of studies reporting on elevated blood pressure, virus-related respiratory problems, mercury exposure and overweight among others, were related to a higher oxidized vs. reduced ratio of this marker.

Similarly, thiol ratios assess the amount of reduced thiol groups (SH) compared to the oxidized form (SS). As opposed to the glutathione ratio, this can be measured extracellularly, representing a marker of predominantly oxidized cysteine residues. There exist several methods to assess the ratio, which may impede comparison across studies. As for glutathione ratio, spectrophotometric measurement including commercial kits are the most common, often based on reaction with DTNB. Stability of thiols and disulphides have been assessed in various samples including plasma and were found to be stable at −80 °C for at least 6 months [194]. Most clinical applications have been originating from Turkey, focussing on CVD [195], respiratory diseases and metabolic disorders [27], as well as cancer, ageing, and neurodegenerative diseases [196], though the relevance for localized diseases such as cancer has been questioned in the same review. The disulphide/native thiol ratio has also been measured in obese children and was elevated compared to non-obese children [197]. The ratio correlated with BMI, as well as with lymphocytes and platelets. No meta-analysis has been reported.

The OSI measures the ratio of total oxidant capacity (TOC) to TAC in a sample. For both measures, a variety of tests can be carried out, such as the oxidation of Fe^2+^ to Fe^3+^ (expressed as µmol/L H_2_O_2_) for TOC, and for TAC e.g., by ABTS (µmol/L Trolox). This measure has been used in a number of small scale studies (mostly in Turkey), including CVD [289] and psychiatric disorders. Major limitations are the various ways to assess OSI, and the lack of interpretability of the final ratio obtained [27], combined with a lack of clinical relevant large-scale studies.

Other indices are based on a large number of parameters, which is clinically impractical, such as the “OS profile”, incorporating 50 measures of OS [290]. While it is physiological meaningful to include various markers into indices, combining enzymatic and non-enzymatic measures, this significantly exceeds clinical practice capabilities.

Finally, the recent emergence of omics- techniques, namely transcriptomics, proteomics and metabolomics, have allowed for gaining additional insights into redox reactions in biological systems, termed “redoxomics” [291]. Though focussing on plant matrices, Ma et al. [292] reviewed the possibility of using -omics techniques for assessing OS, and the major challenges are relevant for human studies as well, namely low number and quality of biomarker validation approaches due to (a) poor characterization of biomarkers, (b) limitation in analytical techniques, (c) difficulty to obtain sensitive and specific biomarkers. In addition, though OS signatures can be revealed by proteomics, metabolomics or transcriptomics, often these techniques do not correlate well, impeding interpretation. Proteomics have been applied to determining e.g., reactive carbonyl species adducts [293], HNE-adducts [294], nitrated proteins [295], or targeting a multitude of oxidative changes, further including methionine sulfoxide, thiol oxidation, and polyubiquitinylation [296]. To a lesser degree, metabolomics approaches have been developed for studying OS [297], revealing altered metabolites including amino acids, ascorbic acid, tocopherols and GSH. Fewer results are available regarding transcriptomics. In a review by Mendiola et al. [298], transcriptomics was employed to study OS in neuroinflammation, revealing the expression of genes involved in coagulation and GHS-related pathways involving microglia and macrophage-infiltration, highlighting the usefulness of the technique [298]. A novel approach employing -omics tools is measuring micro-(small non-coding) RNA’s, which can bind to mRNA, thereby influencing gene expression. However, limited data to disease relevance are available, though susceptibility to RA was shown in a meta-analysis [299], and their potential role for other diseases, such as neurodegenerative ones an the related interplay to OS have been reviewed [300].

Thus, though novel -omics techniques allow for a deeper insight into changes of a broad variety of metabolites, interpretation is still a challenge, and for sure such techniques are not yet widely available and ready to be implemented in clinical practice. The same can be said for OS indices, though e.g., GSH as well as thiol ratios appear physiologically meaningful and have been employed a limited number of clinical studies. MicroRNAs are an interesting and perhaps promising addition to the more conventional markers and have been assumed to be involved in the activity regulation of 50% of protein coding genes [301].

### 4.8. Dietary Indices, Questionnaires

A few approaches exist to determine the effect of dietary antioxidant intake and its relation to OS, as reviewed by Luu et al. [302]. These typically attempt to determine the intake of an array of dietary antioxidants, most often by FFQs. Reported indices include the Dietary Antioxidant Quality Score (DAQS) and the Composite Dietary Antioxidant Index (CDAI). The DAQS displays six scores of the antioxidant capacity sum of the vitamins A, C, E plus Se, Mn and Zn, from very poor (0) to high antioxidant quality of the diet (6). The scores concern the ratio of the daily FFQ-based intake of these nutrients to the corresponding recommended intake. However, this score disregards secondary plant compounds. In addition, the authors investigating the relationship between the DAQS and major OS biomarkers did not report any significant relationship, i.e., with urinary F2-isoprostanes, PGE-2 or other markers of OS or inflammation. The CDAI includes the same nutrients as the DAQS, but uses a different methodology for its calculation, derived from the dietary antioxidant index initially developed by Wright et al. [201]. This latter index combined the consumption of flavonoids, carotenoids, vitamin C and E and selenium. The scoring was constructed based on principal component analysis and summing up of the retained principal component scores. Flavonoids included quercetin, epicatechin, catechin, myricetin and kaempferol. Carotenoids included α-, β- and γ-carotene, as well as β-cryptoxanthin, lutein, zeaxanthin and lycopene. Vitamin E included α- and β-tocotrienol as well as α- and γ-tocopherol. This antioxidant index has been associated with reduced lung cancer risk in male smokers. Rivas et al. [199] previously showed that a diet rich in antioxidants as assessed by the DAQS was positively associated with high bone-mineral density in women. In addition, Farhangi et al. [200] highlighted a significant gene 6P21 rs2010963)-nutrient interaction as assessed by the dietary total antioxidant capacity (DTAC) score previously developed by Rivas et al. [199], as a combination of the vitamins A, C, E and the minerals Se, Mn and Zn. In particular, the 6P21 rs2010963- CC genotype was associated with low DTAC scores and high BMI, blood pressure and FBG [200]. The vitamin and mineral intake were assessed with a semi-quantitative FFQ and was compared to the recommended intake of nutrients. A score equal to 1 was assigned to a vitamin/mineral intake ≥ 2/3 of the recommended intake of nutrients. The DTAC scoring results in a ranking from poor (0) to high (5).

Interestingly, the DTAC indicator developed by Puchau et al. [303] was based on measured antioxidant capacity of food items (FRAP), summing up the individual TAC values of the analyzed food items, taking into account cooking procedures. The authors in particular observed significant correlations between DTAC and the MD score [303]. Likewise, such an approach was used in Poland [304], and a significant relationship was found between the antioxidant capacity of dietary intake, as assessed by the dietary antioxidant index, and an increased antioxidant defence assessed by FRAP as well as a decrease of plasma MDA levels [304].

In addition to FRAP, also ORAC was used to calculate DTAC. For instance, Abshirini et al. [305] assessed dietary intake by a FFQ. The individual dietary antioxidant capacities of each food item determined by the ORAC assay were summed up to determine the DTAC. The so derived DTAC was validated against MDA determined in plasma [305]. Furthermore, the authors showed negative associations between low levels of DTAC, increased serum MDA concentrations and depression/anxiety. This emphasised that high antioxidant capacity and decreased OS can be important not only for physical health, but also for mental wellbeing [305]. However, as with TAC determined in plasma—even less directly—such indices only gather information about antioxidant intake, neglecting host-related biochemical processes of OS.

Finally, for a more comprehensive assessment of TAC in the body, some OS questionnaires with questions comprising food intake, smoking, physical activity, sleeping patterns, drinking and radiation [306] exist, but their validation in a scientific context or use in clinical research has been limited [306]. Such questionnaires have rather been employed in acute OS exposure phases due to a specific environmental context such as benzene or malaria exposure [307,308]. More comprehensive questionnaires and indices taking into consideration a global assessment of the antioxidant capacity and OS, and not only dietary factors should be targeted, especially in the context of non-communicable diseases.

## 5. Markers of Inflammation, Relation to Disease and Practical Aspects

Biomarkers of inflammation are typically serum-, plasma-, or blood-derived proteins or enzymes that have independent diagnostic and prognostic value by representing an underlying state of disease, including cytokines and APPs. Furthermore, biomarkers include counting cell populations related to immune responses, such as neutrophil to lymphocyte ratio (NLR). Additionally, somewhat more secondary markers of inflammation include e.g., COX-2 mediators, i.e., prostaglandins (Table 2). Furthermore, transcription factors, especially NF-κB as a regulator of inflammation have received attention, as have complementary measures, such as DII or more disease specific questionnaires, though with the advantage to be less invasive.

### 5.1. Blood Cell Counting

Leukocyte counts have emerged as a marker of inflammation that is widely available in clinical practice. Complete blood cell counting is an inexpensive and practical test, yielding important information about the patients’ cell populations, and can be carried out by flow cytometry/FACS and related methods. Cell counting should be done on fresh whole blood samples, ideally within 3 h [326]. Especially basophilic counts have shown to change drastically with storage of samples at 4 °C [401]. Freezing, even for a few days at −70 °C can result in significant changes in counted cell populations [327]. Cell counting in the clinic is done by several automated systems, including impedance and optical systems, as well as image cytometers [328]. Elevated levels of almost all subtypes of WBCs, including eosinophils, monocytes, neutrophils, and lymphocytes, as well as NLR and eosinophil–leukocyte ratio constitute independent predictors of inflammation-related diseases [329]. For example, elevated leukocyte counts (>11 × 10^9^/L) resulted from various stimuli such as viral/bacterial infection, with upregulated cellular production by the bone marrow [402]. Elevated counts have also been associated with worse outcomes in coronary diseases, in a meta-analysis related to MetS [330], with T2D (including granulocytes and lymphocytes but not monocytes) [331], and even in the general population [329]. A low lymphocyte count (lymphophenia) can also signify a reduced capability of the body to fight infection. Low lymphocyte counts were found in meta-analyses of severe COVID-19 [332], and can be used in HIV infections as a decision-aid for starting anti-viral therapy [333]. These cells may be a target of infiltration by certain viruses, or may be a result of an IL-6 reduced production [332].

Regarding eosinophils, participating in cellular immune responses, their normal range is 0–0.5 × 10^9^/L [403]. During acute infection, eosinophil levels drop rapidly. A complete absence of eosinophils is seen in serious infections including sepsis [404]. Elevated levels of eosinophils (eosinophilia) can be harmful as eosinophils contain granules rich in biologically active molecules such as cytotoxic proteins meant to kill pathogens. These can damage tissues when present at large quantities [405]. Eosinophils increase in allergic diseases such as asthma as shown in a meta-analysis [334] and also in several autoimmune diseases, such as celiac disease [335], and in IBD (both CD and UC) [336]. In some forms of cancer, high levels of eosinophils are symptomatic, as in Hodgkin’s lymphoma [337].

Monocytosis (high levels of circulating monocytes, important for the innate immune system), may increase counts to >0.8 × 10^9^/L. This condition is frequently associated with infections such as malaria [338], autoimmune diseases such as rheumatoid arthritis [339], IBD and colorectal cancer as shown in a meta-analysis [340], obesity [341], and schizophrenia as also found in a meta-analysis [342], while no elevation was seen in a meta-analysis of T2D [331].

Most functions of basophils depend on the release of heparin and histamine [406]. Basophils contribute to only 1% of the total white blood cell count, their number normally being <0.20 × 10^9^/L [343]. An elevated level (>2.0 × 10^9^/L, termed basophilia) has been associated with autoimmune inflammation or allergies [344]. For example, a low level of basophils (basopenia) may be associated with RA [345].

Not only leukocyte cell counts but also the ratio between these subpopulations and even platelets may present promising biomarkers. For instance, in a case–control study of 190 individuals with/without CAD, the number of eosinophils and also the eosinophil to leukocyte ratio in peripheral blood were significantly elevated [407]. A high NLR can be a predictor of heart disease. A high NLR was also reported for poor glycemic control in T2D [346]. Multiple studies have shown that increased NLR is associated with poor prognosis in a variety of cancers, such as colon, pancreatic, stomach, and lung cancer. In a meta-analysis of >100 studies and 40,000 patients, an NLR>4 was associated with worse solid tumor survival rates [347]. Finally, lymphocyte-to-monocyte ratio (LMR) correlated with diverse malignancies and CVD. LMR possesses good predictive value for the prognosis of acute ischemic stroke and lower LMR was related to with stroke severity and poor outcome in a previous meta-analysis [348]. Low LMR was also significantly associated in a meta-analysis with lower overall survival in patients with ovarian cancer [349] and with solid tumors according to a meta-analysis of 29 studies [350].

To obtain improved reference values for cell-count-based markers of inflammation, >8000 elderly were investigated in the Rotterdam study [408]. In this population, means (95% CI) for NLR and platelet-lymphocyte ratio were 1.76 (0.83–3.92) and 120 (61–239), respectively. A systemic immune-inflammation index (SII) was also evaluated, incorporating peripheral lymphocyte, neutrophil and platelet counts. In a meta-analysis of 22 articles and >7500 patients, SII was proposed as a good predictor for tumor progression and survival in several cancers [351].

Taken together, cell-counting methods offer valuable insights into a variety of inflammation-related diseases and appears to be related with the prediction of a variety of diseases. However, acute infection may also significantly affect differential leukocyte cell counting. A rapid response in counts of WBC has been reported, such as 12 h following operation [409]. As WBCs are at the origin of a large number of cytokines, triggering inflammation, cell counts offer a good proximate marker of ongoing inflammation, but analyses must be conducted rapidly from fresh blood samples, somewhat limiting the practicality of these measures.

### 5.2. Cytokines/Chemokines

Although useful as first appearing markers of inflammation (within a few hours after e.g., injury), their half-life in blood in vivo is short, a few minutes for most [410], and measured concentrations may vary rapidly with time. Also, their stability in blood plasma, even when stored at −80 °C, is limited. While 1 year of storage did not compromise the stability of several cytokines (IL-1β, IFγ, IL-6, TNF-α), degradation was obvious at year 2–3 [310]. Storage stability was stated as IL1β>IFα>IL-1α>IFγ>IL-6>TNF-α when spiked to plasma upon freeze-thaw cycles [411], though decent stability for 3 freeze-thaw cycles was reported for IL-6, IL-10, IFγ, and IL-2 [309]. Regarding detection, cytokines can be measured by immunological methods (e.g., ELISA), though simultaneous measurements via e.g., multiplex bead array assay methods are superior regarding sample throughput [311].

IL-6 production is up-regulated during acute infection. It is also produced by adipocytes and higher levels in individuals with obesity have been reported. In line with this, in a meta-analysis, IL-6 polymorphisms were related to lower risk of obesity [315]. Diseases associated with IL-6 are numerous. In a meta-analysis of 8 cohort studies, cognitive decline was related to increased IL-6 concentrations in elderly without dementia [312]. IL-6, in a recent meta-analysis of 8 studies on COVID-19 was 3-fold elevated in patients with vs. without complications [313]. In another meta-analysis (four studies), subjects with irritable bowel syndrome exhibited elevated IL-6 levels [314]. Elevated circulating levels of IL-8, as IL-6, have been related to a number of chronic diseases. These included, according to a meta-analysis of 16 cross-sectional/case-control studies individuals with lupus erythematodes [316], hepato-cellular carcinoma in a meta-analysis of 10 studies [317] and periodontitis according to a review of IL-8 polymporphisms [318]. No relation to obesity was found in a meta-analysis based on 34 articles [319].

TNF-α polymorphism was related to colorectal cancer in a meta-analysis of 22 studies [412] and increased TNF-α (also IL-6 and 8) was found in a meta-analysis of breast-cancer [325]. Silva et al. have further highlighted the role of TNF-α in numerous diseases from Alzheimer’s to rheumatoid arthritis, IBD, and several types of cancer [413] Individual studies also suggested higher circulating levels in persons with obesity [323]. However, in a meta-analysis of 5 prospective studies it failed to show a relation with T2D, unlike IL-6 and CRP [324].

The polymorphism of this IFγ has been related in a meta-analysis of nine case-control studies to breast cancer [320] and in a meta-analysis of eight studies to cervical cancer [321]. Interestingly, IFγ levels were not significantly associated with mycoplasma-caused pneumonia in a meta-analysis of 6 small-medium scale studies [322].

Neopterin, produced by macrophages, is stable in the human body but secreted via the kidney, with a half-life of 90 min. [414]. It is a ROS quencher aiding in the protection of macrophages [415]. Its stability in plasma has been reported as 3 d at RT or 3 months at −20 °C [111]. Neopterin has also been related to a number of diseases. Though no meta-analysis has been reported on, plasma levels were related to homocysteine and risk of CVD [388], and as well as to cancer such as ovarian cancer [389]. It was also shown to be modulated by age, gender, and BMI [390].

To conclude, cytokines are the first line of defence and primary formed agents of inflammation, and thus valid markers of inflammatory processes, further triggering activation of a number of transcription factors and APP, enhancing inflammation. Some cytokines may play a dominant role in cancer (such as IFγ), while others such as IL-6 may also exhibit anti-inflammatory properties and may not be suitable on their own to generally assess systemic inflammation. A limitation is further their short half-life in circulation and limited storage stability.

### 5.3. Acute Phase Proteins and Acute Phase Reactants (APR)

Traditionally, CRP has been used to assess inflammation levels in the body and to monitor chronic health issues such as autoimmune disorders. Recent work has further indicated that mental disorders, including bipolar disorder and chronic depression, are often linked with abnormalities in the immune system, including CRP [416]. For example, it was demonstrated that CRP levels can predict whether a patient with a non-bipolar major depression may respond better to medication or behavioural intervention [417]. Whether treatment of the immune system can positively affect mental disorders remains to be further elucidated. In line with such findings, in a recent meta-analysis, higher levels of CRP in the blood were found in Parkinson’s disease patients compared to controls [354]. There is also confirmation for CRP being a risk predictor of CAD [418]. Mortality from stroke and CHD were likewise positively related to circulating CRP [355]. CRP is also detectable shortly after acute insult, such as after operation, within 48 h [409], thus trailing somewhat behind the cytokine response. Regarding its measurement, it appears that CRP levels are fairly stable, which contributes to its rather frequent measurement. During 11 years of storage at −80 °C, no significant changes in serum CRP concentrations were encountered [352]. Often, CRP is assessed by ELISA, though in clinics automated measures such as by immunoturbidimetry are employed [419].

Due to the involvement of SAA in atherogenic processes, Mozes et al. concluded that damage to the myocardium is among the most powerful stimulus for SAA induction, followed by traumatic events, arthritis, viral infections, and malignant diseases [420]. Meta-analyses have further revealed a strong relation between elevated SAA and obesity [14], CHD [15], poor overall survival in individuals with solid tumors [360] and worse outcomes in lung cancer [361]. Various assays for SAA such as radioimmunoassay, radial immunodiffusion, and ELISA tests have been reported, though these are mostly used in research laboratories. In clinical practice, automated latex agglutination immunoassay are typically used [359]. Stability of SAA has been reported as at least 17 d at −20 °C [358], no report on storability at −80 °C was found.

Reduced haptoglobin levels were observed during liver impairment, malnutrition, and congenital hypohaptoglobinemia. Haptoglobin also increases in inflammatory diseases, in cigarette smokers, during nephrotic syndrome [366], in rheumatoid arthritis [367]. Marchand et al. showed that a haptoglobin limit of <25 mg/dL allowed identifying hemolytic from non-hemolytic disorders [421]. In a recent meta-analysis, haptoglobin genotype was associated with cardiovascular outcomes [365], something also observed in an earlier proteomic study of heart failure [422]. For its measurement, common tools include spectrophotometry [423], immune based methods and gel electrophoresis. Nephelometric determination in plasma is based on automated laboratory systems [364]. Stability of haptoglobin in bovine plasma was at least 3 weeks at −20 °C and −80 °C [362] and 120 d for haptoglobin in saliva stored at −20 °C [363].

It was shown that MBL insufficiency predisposed to higher severity and fatal outcome in patients with pneumonia [424]. In a meta-analysis of MBL and polymorphisms, significant associations were found with sepsis [369] and vulvovaginal infections [370], but not with hepatitis B (though severity) [371]. In addition to infectious diseases, it was shown that MBL was elevated in patients with systemic lupus erythematosus [425]. Decreased concentrations of MBL were also associated with COPD [426]. The most used detection method is ELISA. Pure protein can be stored for 1 year at −80 °C [368], no data on plasma is available.

Decreased serum albumin was associated with reduced kidney function and poor renal prognosis in patients with T2D and diabetic nephropathy [427]. Decreased serum albumin is also present in chronic IBD [428]. However, albumin is not specific for inflammation, and may indicate a low-protein diet [429], though intake of sufficient macro-and micro-nutrients does not fully protect from hypoalbuminemia during physiological stress such as trauma or infection [125].

In elderly, TTR was found to be inversely related to inflammation (measured by CRP) [430]. The authors have highlighted that TTR well correlated to ROS/RNS, and its gene expression is regulated by stress hormones, among other. Decreased levels have been associated with mortality in acute kidney injury [431], though more typically, low serum TTR is often encountered in patients with severe protein-calorie malnutrition, and thus seen primarily as a marker of nutritional protein status. Both albumin and prealbumin are assessed by immunological methods such as immunoturbidimetric and immune-nephelometric assays. Samples have been reported to be fairly stable during 1 year storage at −80 °C [432].

Transferrin has been associated with certain diseases such as decreased levels in IBD [433]. In urine, it has also been regarded as a potential biomarker for kidney damage such as following T2D [434], constituting a more indirect marker of a number of chronic diseases. However, it is strongly influence by iron status and result should be interpreted with care.

Procalcitonin has been discussed as a sepsis marker to differentiate bacterial from viral diseases (being rather downregulated in the latter case), though this was not confirmed in a recent meta-analysis of pneumonia [373]. However, another meta-analysis to differentiate between IBD and infection found that it was significantly elevated in the latter [374]. This marker is said to be present in acute infection several hours after the start of inflammation (3–6 h), and may stay elevated for 1–2 days [435]. In addition to infection, it has been reported to be elevated in persons with obesity, where it also correlated significantly with hs-CRP [436]. Thus, procalcitonin may constitute a marker for bacterial infection and related inflammation, rather than a general marker of inflammation. It is measured frequently via immunological techniques (ELISA), and storage at −80 °C caused only minor losses [372].

In summary, especially CRP and SAA have been frequently measured as APPs. They are reasonably stable and rather established markers of inflammation, and both have been associated with several diseases. Other APPs such as albumin, prealbumin and transferrin lack specificity to be suitable inflammation markers, and may be influenced also by nutritional status, while procalcitonin may be a marker for bacterial infections. For haptoglobin and MBL, both may be related to inflammation and disease, but compared to CRP and SAS, data is scant.

### 5.4. COX-2, Endothelial Markers

While it is possible to measure COX-2 (e.g., by ELISA), due its transient up-regulation, it is more common to assess downstream prostaglandin products. Increased PGE-2 levels have been associated with several chronic diseases. For instance, PGE-2 was elevated in women with obesity [380], and especially in patients with cancer, including colorectal [381] and ovarian cancer [382]. Also T2D patients showed elevated concentrations [383], and its contribution to a variety of neurological disorders has been proposed [384]. It also appears to be rapidly elevated following stimuli. In a cat model of ischemia, 15 min. after injury. PGE-2 levels were significantly elevated, with a peak at 60 min. [437]. Most methods to measure PGE-2 concentrations in serum/plasma employ immunological methods for its detection such as ELISA. Also mRNA detection via PCR and MS-MS based methods have been used, avoiding the risk of cross-reactivity [438]. The stability of PGE-2 in plasma was specified as only a few days in saline and possibly also plasma at 4 °C [439], data on storability at −80 °C appears unavailable, but can be inferred to be limited to weeks, as PGE-2 will degrade to other prostaglandins (PGA, PGB).

Another group of potential markers of inflammation which deserve attention are those of endothelial inflammation, as these may be related to CVD such as atherosclerosis. Over two dozen endothelial markers with a potential relation to inflammation have been reported [440], though intercellular adhesion molecule 1 (ICAM-1 or CD54) and vascular adhesion molecule 1 (VCAM-1 or CD106) are the most investigated ones. Both can be stimulated by certain cytokines, including TNF-α. ICAM-1, a transmembrane glycoprotein, can bind leukocytes and enables their further migration to peripheral tissues, increasing endothelial permeability. In a recent meta-analysis, a polymorphism of a gene expressing ICAM-1 was related to risk of CAD, highlighting the importance of this marker for vessel-related health [441]. VCAM-1 is expressed at cellular surfaces, facilitating attachment of lymphocytes, monocytes and other cells to the endothelium. As ICAM-1, it is upregulated by cytokines. Of note, in serum/plasma the soluble (s) forms of ICAM-1 and VCAM-1 are measured, which are thought to reflect the bound fraction, as they appear to origin from the cell-membrane bound form, following cleavage by a metalloprotease to produce soluble ICAM-1 [442]; similar cleavage is thought to result in soluble VCAM-1 [443]. In a meta-analysis of 15 prospective studies on T2D, increased circulating ICAM-1 but not VCAM-1 was related to elevated risk of T2D [375]. In a meta-analysis of 18 case-control studies, ICAM-1 polymorphism was linked to cancer risk [376], similar to a meta-analysis of 12 case-control studies on T2D [377] and CHD in a meta-analysis of 11 case-control studies [378]. On the other hand, VCAM-1, but not ICAM-1, was associated with preeclampsia in a meta-analysis of 21 studies [379]. Thus, ICAM-1 and VCAM-1 may be good prognostic markers of endothelial vessel health, e.g., atherosclerosis and CVD outcomes, but are rather secondary markers compared to cytokines, further downstream in the inflammation cascade. Thus, their prognostic value for other diseases is less clear, but the emerging role of VCAM-1 in several neurological disorders and cancers has been emphasized [444], as has the role of ICAM-1 in psychiatric disorders and peripheral immune system [445]. Soluble ICAM-1 and VCAM-1 can be measured in plasma, e.g., by ELISA. As for PGE-2, there appears to be a rapid increase in sICAM-1 and sVCAM1 following injury. In patients with basal ganglia haemorrhage, levels were highest in the first hours after injury [446]. Their stability in lyophilized forms in ELISA kits is given with several months at −20 °C, no results for long-term storage have been published. Taken together, while COX-2 and downstream prostaglandins such as PGE-2 constitute fairly established markers related to inflammation, the roles of ICAM-1 and VCAM-1 in inflammatory processes are still emerging.

### 5.5. Transcription Factors

The advantage of measuring NF-κB is to capture a broader overview of pro-inflammatory status. However, there are also disadvantages. For NF-κB, only the fraction that would translocate to the nucleus would be able to act in a pro-inflammatory fashion. Secondly, as for all inducible transcription factors, NF-κB stimulation, similar as to Nrf2, may only persist for a limited amount of time, and measurements may be more time-sensitive than the resulting downstream cytokines. This was shown in a rat model of cerebral haemorrhage, where NF-κB protein expression was measured 3 h following injury (cytoplasm), identified in the nucleus at 6 h, with a peak expression of 72 h, lasting for about 5 days [447]. Finally, expression may differ between cells in the body and measuring NF-κB may require a fair amount of cells. This could be achieved by either ELISA (about 10–50 µL of a protein concentration of 10 µg/mL or a total of 0.1 µg absolute protein amount; when working with individual cells, ca. 10^7^ cells/mL are often targeted [385]), or more sophisticated transcriptomics [448]. Its stability is limited, but cellular extracts may be stored for 2 months at −80 °C [385].

Despite these limitations, NF-κB has been measured in inflammation. Expression can be assessed via antibodies, and NF-kB p65 and NF-kB p50 are the most common ones. In a previous meta-analysis, NF-κB expression was related to worse tumor prognosis and survival [25], and this was true for both cytosolic and nuclear NF-κB, which would circumvent the need to prepare nuclear extracts. A similar relation was found in another meta-analysis of non-small cell lung cancer and NF-κB expression [386], but only those studies investigating nuclear vs. cytoplasmic levels found a positive relation. Though appropriate human studies appear to be missing, studies in rodents have shown that NF-κB is activated in high-fat fed animals and in obese ones [449]. In isolated mycocytes from individuals with obesity, elevated NF-κB was also shown [387].

STAT1 has been related to prognosis of early-stage colorectal cancer [450], as constitutively active STAT is a hallmark of many cancers, also correlating inversely with FBG levels in obese humans and mice [451]. Also other inflammatory diseases, especially IBD, have been related to STAT3 activation [452]. Other transcription factors which are involved in inflammation include peroxisome proliferator-activated receptors (PPARs), involved in obesity/adipocyte differentiation [453] and the MetS [454], as well as lipid signalling in cancer [455], and MAPK, related to e.g., inflammatory processes in the brain [456], but less data is available on these regarding inflammation related diseases, and may on their own not represent a suitable marker.

In summary, NF-κB is an interesting marker of inflammation, involved in a multitude of gene expressions with pro-inflammatory responses. Impeding its clinical use are the various techniques and antibodies existing to measure this transcription factor, as well as the potential requirement of isolating the nuclear fraction. Finally, expression may differ depending on time and cell type.

### 5.6. Composite Markers, Indices, Omics

Like for OS, composite markers and indices have been developed for inflammation assessment. An HDL inflammatory index (HII) was created based on isolated HDL (e.g., by precipitating LDL with dextran sulfate) particles of individuals to protect against likewise isolated “standard” LDL particle-oxidation, measuring dichlorofluorescein photometrically over time [457]. Thus, it is rather a measure of oxidative stability than inflammation, however, the name was derived from the relation between endothelial inflammation and the increased risk of atherosclerosis with instable lipoprotein particles. Disadvantages include the time-consuming measurements as well as the need of an LDL standard. Though no meta-analyses were conducted, increased HII has been correlated to sepsis and shock in individuals with organ failure [395], poor outcome in subjects with hemodialysis [396] and diabetes [397], among others.

A composite score based on five inflammatory markers was proposed by Walker et al. [458]. In their study relating inflammation to late-life brain volume, inflammation was assessed by APPs albumin, fibrinogen, factor VII, van Willebrand factor (a factor preventing bleeding, but also being associated with several diseases including CVD) and leukocyte count, by averaging standardized z-scores. Individuals with 3 or more individual elevated markers showed a mean reduction of hippocampal volume by 5%, proposing that such measures may be early markers for cognitive aspects during ageing. Another score with rankings from 0–5 was proposed earlier [28], based on leukocyte count, erythrocyte sedimentation rate, CRP, and soluble receptors 1 and 2 of TNF-α. The authors found a correlation to basal insulin resistance surrogate indices. Bonaccio et al. used a score composed of CRP, leukocyte and platelet counts, as well as granulocyte/lymphocyte ratio (Section 5.1), which was related to T2D mortality in >20,00 participants [459]. However, such composite scores or indices have only been used by individual teams, no generally accepted composite score or index appears to exist.

A novel marker, a proton signal assessed by NMR, is GlycA [460]. It has been recently proposed as a marker of chronic inflammation, predicting mortality [392]. This marker detects the N-acetylmethyl group protons of N-acetylglucosamine residues and thus glycosylation state on glycan associated proteins, including APPs. Glycosylation is a frequent post-translational modification of secreted proteins, and altered glycan structures of circulating proteins have been recognized as markers of various diseases [461]. GlycA was related to individual inflammatory markers such as TNF-α, CRP, IL-6 [460], and glycosylated immunoglobulins were associated with chronic diseases such as RA, lupus erythematosus and HIV, but also of acute febrile illnesses [462]. The strong relation to total mortality and cancer [392] makes this an interesting marker, which was inversely associated with exercise and visceral fat [393]. However, it requires less frequently available instrumentation, and potential difficulties of interpretation due to overlapping signals were mentioned [460]. Regarding stability, as glycosylation rather enhances the stability of proteins [391], mid-term storage at −80 °C is assumed not to impact sample measurement.

Regarding -omics based approaches, in a database-search by Saha et al. [463], a set of 204 plasma proteins related to inflammation processes were retained, allowing for more in-depth insights in involved cytokines, various enzymes, nuclear factors, and many more. Otherwise, most -omics studies rather followed a disease driven approach to reveal differentially regulated proteins in healthy vs. diseased subjects. Such investigations revealed proteins involved in inflammatory and lipid homeostasis following gastric-bypass surgery [464], arthritis [465], visceral adipose tissue between subjects with obesity and of normal-weight [466], among others. Likewise, in metabolomics, children with vs. without appendicitis were differentiated by a set of nine metabolites and seven inflammatory markers [467], as were children with IBD [468]. Metabolomics was further used to detect low-grade chronic inflammation in apparently healthy individuals [469], and prognosis of inflammation state based on the 10 most strongly differentially regulated markers was feasible, including phospholipids and the novel oligosaccharide marker 3′-sialyllactose, shown to have immune-modulation properties in neonates. Finally, transcriptomics was similarly employed to detect signatures of inflammation in subjects with myocarditis, revealing a final set of 13 genes [469], and systemic inflammation in sepsis with a final set of 7 genes [470], including immune-related genes such as for toll-like receptor 5, clusterin, fibrinogen-like 2, and IL-7. As for OS, these techniques allow for much broader and fundamental insights, but are unlikely to be employed in regular clinical facilities anytime soon.

### 5.7. Questionnaires

The dietary inflammatory index (DII) was developed in 2009 to assess the inflammatory potential of diets [471]. This index relates to overall dietary pattern, summing up individual food items as either pro-or anti-inflammatory, based on literature data relating these items to various inflammatory markers (cytokines, CRP). Points were given for both pro-and anti-inflammatory findings, which were further weighted according to the type of study—with a higher weight for human than for in vitro studies. The DII was updated in 2014 [472], including additional literature data and a total of 45 food groups, and has been related to a number of chronic conditions, in addition to CVDs. For instance, DII was related in meta-analyses to cardiovascular mortality, all-cause mortality [398], cancer incidence [400] as well as obesity occurrence [399]. Disadvantages are the need for a complete FFQ assessment, being time consuming and prone to errors due to inaccurate intake estimates. Underreporting for example is a frequent limitation of food-based questionnaires. Furthermore, it only relates to certain anti/pro-inflammatory compounds of the diet, such as fiber, ω-6 fatty acids, polyphenols, carotenoids, sugars, saturated fats etc. Thus, it may overlook other pro/anti-inflammatory dietary constituents and neglect other sources of inflammation. The anti-inflammatory diet index (AIDI) was developed to predict low-grade chronic inflammation in the Nordic population, using a 123 item FFQ [473], while the EDII (Empirical DII) is more specific to the US population [474]. Both indices have found less application than the DII. Additional questionnaires related to inflammation exist, mainly targeting inflammatory bowel diseases (IBD) [29]. Some questionnaires targeting chronic pain also exists, such as the chronic pain grade questionnaire, which may serve as an approximate estimation of inflammation [475], but was not yet validated against biological inflammation markers. A diverse number of online questionnaires also exist, likewise non-validated.

## 6. Conclusions and Perspectives

A large array of markers related to OS and inflammation have been reported in the literature, and it is challenging for researchers or clinicians to keep a sound overview on the topic and choose the most appropriate one (Figure 3).

Due to the interrelation between many of the processes related to OS and inflammation, several of the reported markers may be elevated at the same time. It is even advisable to assess more than one single marker in order to obtain a more complete picture on the ongoing processes involved, e.g., ideally assessing two complementary markers of both OS and inflammation. Though correlations between markers may be somewhat disease and also time specific, a few studies have reported correlations of OS and inflammatory markers, either between each category or across.

For instance, in a study involving Tunisian coronary artery disease patients (*n* = 120), a positive significant correlation between total antioxidant status in plasma and hsCRP was observed in those having diabetes. The increase in antioxidant capacity appeared to be driven by the elevated glucose concentrations [476]. In both patient groups, total antioxidant capacity, GPx and SOD, measured in erythrocytes, were positively and significantly correlated, though all correlations found were very low (r < 0.1). As the latter 2 may both be related to intracellular hydrogen peroxide metabolism, the correlation is not too surprising. Similar results were obtained regarding the relation of SOD and GPx in ischemic stroke patients [477]. In another study with patients with coronary artery disease (*n* = 385 individuals), the association between hsCRP, MDA, erythrocyte SOD and O_2_^•−^ concentrations were examined. MDA correlated significantly with hsCRP and erythrocyte SOD, but not with O_2_^•−^ (the correlation between the latter two did not reach statistical significance), highlighting the relevance of MDA as a marker in coronary disease, as reviewed by the authors. CRP was also positively (and significantly) correlated in patients with heart failure, to SOD and MDA [478]. In patients with chronic obstructive pulmonary disease [479], sICAM-1 was significantly associated with IL-8, though inversely with GPx and TEAC (the latter two being positively correlated), also emphasizing the interrelation of inflammatory and oxidative stress markers. In patients with high risk of CVD (*n* = 60), levels of CRP, IL-6 and F2α-isoprostane (in urine) were investigated [480]. While the measured isoprostane correlated significantly with CRP, it did not with IL-6, perhaps due to a more time-dependent appearance of cytokines in inflammation, though this is speculative. However, the study emphasised the interrelation of OS and inflammation, at least for CVD. Also, as perhaps expected, due to the similar underlying primary causes of lipid and protein oxidation, AOPPs correlated strongly (R = 0.5) and significantly with MDA in patients with Crohn’s disease [176]. It can be concluded that several markers of inflammation and oxidative stress may be positively correlated with one another, though markers of antioxidant activity (SOD, GPx) and antioxidant capacity (TEAC etc.) may either be upregulated as a reaction to OS, or may be deficient in certain patients, pointing to deficiencies in coping with OS, thus being either positively or inversely correlated to OS/inflammation. For example, in a study with hyperuricemic individuals [481], SOD was inversely related to IL-6 and TNF-α, while MDA was positively related to both.

In the presence of rather acute stimuli, a certain time-dependency for the appearance of markers of inflammation and OS can be assumed (Figure 4), owing that the biological cascade from stimuli → first cellular reactions and products (ROS, cytokines) → humoral compartment (APPs), WBCs, macromolecule (per)oxidation → transcription factors → antioxidants/antioxidant enzyme regulation → late oxidation products (AOPPs), tissue damage etc., will take time.

For instance, in a study on mice exposed to hyperoxia [482], a first inflammatory response was measured within 12 h (increased IL-6) and 24 h (increased TNFα). This was followed by changes in WBCs between 24 and 48 h. GSH/GSSG ratios were modified within 12 h, CAT only at 48 h after insult, and histological changes were visible between 24–48 h. Surely, such times give only a relative indication and are expected to change according to the animal model and the nature of the insult, among other. Other studies have reported somewhat faster reactions. For example, in relation to high doses of ethanol in a rat study, plasma GR decreased 2–4 h after insult, SOD and GPx significantly increased 4–6 h after insult. The ratio of oxidized to reduced glutathione increased 1.5–6 h after ethanol intoxication. In line with this, also MDA concentrations increased, with a strong increase 1.5–2 h, peaking at 4 h [483]. Therefore, when assessing markers of inflammation and OS in studies, care should be taken to assess these at similar time points, at least in rather acute disease, or even taking repeated measures.

Therefore, there is a large array of markers to choose from. A first level of classification of markers may be their distinction into direct (i.e., original, primary) and more indirect (i.e., secondary) markers. Original markers of OS are ROS, including O_2_^•−^, NO and peroxinitrite. Due to their short-lived nature, their measurement is challenging, but it can be attempted by EPR. However, they rapidly react with lipids, proteins, DNA/RNA to produce secondary markers of OS. Some of these are more stable and allow for a more reliable measurement in adequately stored (i.e., frozen) samples. Such markers include e.g., PCOOHs, MDA, F2-isoprostanes or oxLDL for oxidized lipids, DNA tails or 8-OH-dG for DNA breakdown, or PCs for modified proteins. Further downstream are additional markers related to the consequences of ROS, including the body’s antioxidant system, which may or may not mirror the balance between pro-oxidants and antioxidants. These include antioxidant enzymes (SOD, CAT, GPx etc.), transcription factors (Nrf2), endogenous (e.g., albumin) and exogenous antioxidants (e.g., vitamin C and E), and TAC, which tries to measure both endogenous and exogenous antioxidants. While these latter markers are no direct measure of OS, they may complete the picture of OS homeostasis, as long as a marker of ROS is included. This can include measuring proposed OS indices or composite markers, e.g., glutathione or thiol ratios or the OSI. However, literature on employing these markers are scanter and measuring these indices is generally more labor-intensive.

In summary, no universal consensus exists regarding the most suitable markers of OS, though some such as the more indirect ones have obvious limitations. In order to grant health claims, EFSA recommended the measurement of certain markers with specific methods such as MS-MS, including tyrosine nitration products for proteins, F2-isoprostanes or phosphatidyl-hydroperoxides as a marker for lipid damage, and the modified COMET assay for DNA breakdown. OxLDL by immunological measures was also recommended. These markers have also been shown in meta-analyses to be associated with a number of cardiometabolic and also cancer diseases, though due to practical reasons often immuno-assays rather than the more specific methods such as MS-MS have been used. In addition, it is generally suggested to measure more than one marker of OS, preferably complementary ones, such as F2-isoprostanes for lipid oxidation plus the COMET assay for DNA degradation. This recommendation is likely difficult to be implemented in clinical practice anytime soon, due to equipment and laboratory limitations, though measuring oxLDL together with e.g., nitro-albumin by immunological methods may be more feasible and automatable alternatives for clinical background.

Regarding inflammation, cytokines produced by macrophages and other sensitized cells may constitute the most employed markers of primary inflammation agents, including e.g., IL-1, IL-6, IFγ and TNF-α. Measuring sub-populations of leukocytes or ratios of blood cells may also constitute appropriate markers, and is more available than cytokine measurements, however, they may be more effected by acute viral and bacterial infections. Secondary markers of inflammation, in response to cytokines, such as APPs, mostly CRP and SAA, have been widely used, and are clinically more easily measurable and more stable than cytokines, and thus more data regarding disease associations exist. Some APRs such as procalcitonin appear more specific for bacterial infection and are unsuitable as a general marker of inflammation. Following stimulation by cytokines, also COX-2 products such as PGE-2 and endothelial related VCAM-1 and ICAM-1 may be complementary markers. However, as these are rather consequences of cytokine stimulation, they may be more specific for certain diseases, such as atherosclerosis, not reflecting systemic inflammation. Another, interesting approach is measuring transcription factors involved in cytokine production, specifically the pro-inflammatory NF-κB, which may be also further activated by cytokines, being both cause and consequence of enhanced cytokine levels. As NF-κB is related to many downstream inflammation-related targets, this may yield a more integrative marker than single downstream components such as single cytokine. On the down-side, its levels may be time and tissue dependent, and may require isolation of the nuclear cellular fraction, requiring e.g., leucocytes. Due to these limitations, there is limited data available on transcription factors as markers of inflammation. The same can be said for Nrf2 as a marker of OS, and no absolute reference values have so far been provided.

Laudable, in part more novel approaches include inflammatory and OS indices, though due to analytical challenges and being more laborious, it is unlikely that they will soon be used in clinical practice. Regarding inflammation, an interesting novel approach is GlycA, which has some potential, though at present too little data is available and the requirement of NMR technology severely limits its clinical usefulness. Complementary markers not relying on biological measurement exist, including disease specific questionnaires, or the DII, assessing dietary contribution to inflammation. Such methods can yield non-invasive interesting and at least complementary information. However, considered alone, such indicators are unreliable for assessing inflammatory status, as only a limited aspect of inflammation is investigated, and inter-individual host-differences impede interpretation for the individual. Taken together, as for OS, decent insights into inflammation may best be obtained by combining two complementary markers, e.g., combining cytokines with APPs or blood cell counts.

As a short practical guide, the following recommendations may be considered:The marker chosen should have been shown in previous studies to be related to the underlying cause (e.g., disease) of the OS/inflammatory stimulus (see e.g., tables),More than 1 marker of each OS and inflammation should be assessed,Markers should measure complementary aspects, e.g., a cytokine and APP for inflammation, and for OS e.g., a marker of lipid oxidation (F2-isoprostane) and of DNA degradation (e.g., COMET assay),The most specific and selective analysis available should be employed, e.g., chromatographic techniques should be preferred over immune-related techniques (ELISA),The time-dependency of markers should be considered in acute inflammation/OS, especially for transcription factors. Preferably, multiple measures should be carried out, or at least individuals be measured at the same state/time following insult,Certain markers are more indirect and complementary, i.e., biological responses to OS and inflammation, including antioxidant enzymes and compounds, and should be considered only as additional markers,For samples that have been stored for prolonged periods of time (>1 y at −80 °C), the choice becomes more limited. 8-OH-dG and AOPPs for OS and APPs for inflammation may be good options in such cases.

In closing, the field of OS/inflammation is complex and intertwined with many biological reactions, which are eventually related to chronic diseases. They may constitute both cause and consequence of disease, aggravating one another. No single universal marker of OS or inflammation exists that is sufficiently predictive for all inflammation/OS related diseases. From a disease viewpoint, specific markers such as endothelial damage or oxLDL for atherosclerosis may be preferred over more general markers. The most universal markers, linked to the largest variety of diseases are the more “upstream”, i.e., direct ones, including measures of ROS and protein-, lipid-, and DNA- oxidative damage as well as leucocyte populations, cytokines and APPs as main drivers of inflammation. A number of interesting and novel markers await their clinical employment, and the wider availability of NMR, MS-MS and -omics techniques will allow for more accurate and deeper insights into the relation between inflammation, OS and disease.

## Figures and Tables

**Figure 1 antioxidants-10-00414-f001:**
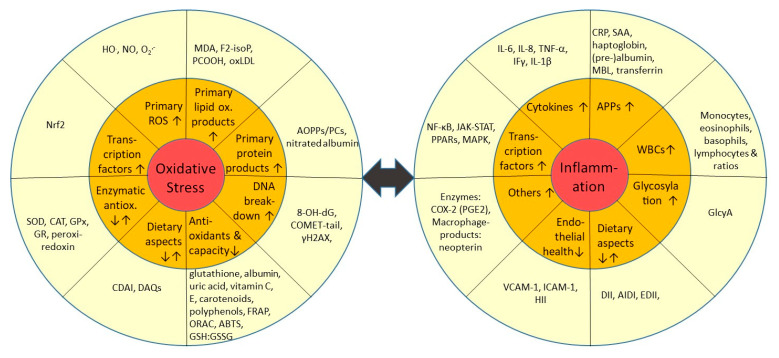
Hallmarks of OS and inflammation and altered parameters that can be assessed to determine their status.

**Figure 2 antioxidants-10-00414-f002:**
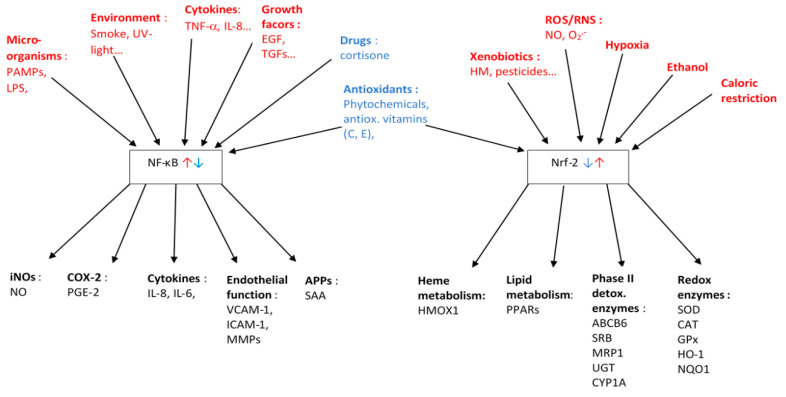
Major relation between NF-kB, Nrf-2, and downstream genes related to inflammation and OS.

**Figure 3 antioxidants-10-00414-f003:**
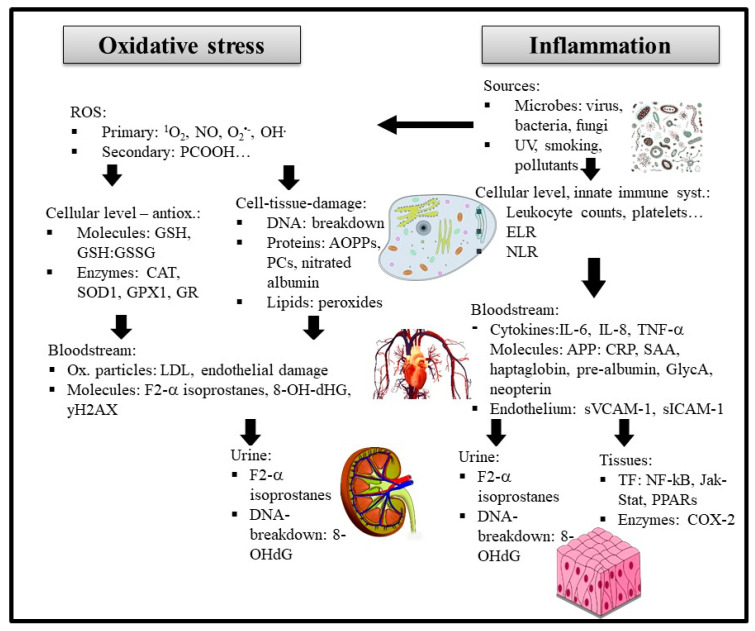
OS and inflammation processes and markers involved in respective body compartments.

**Figure 4 antioxidants-10-00414-f004:**
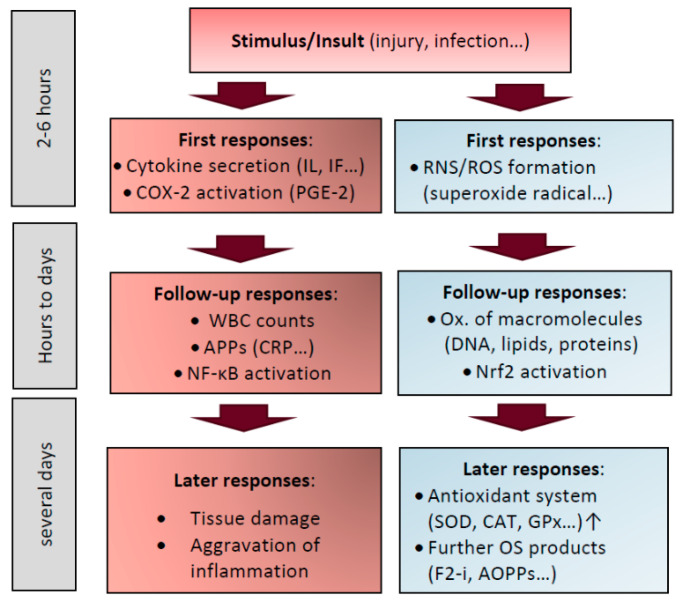
Proposed time-dependency of markers of OS and inflammation following an acute insult.

**Table 1 antioxidants-10-00414-t001:** Overview of common markers of OS—advantages and disadvantages.

Marker	Class of Marker	Measured in: Matrix	Stability	Techniques Used	Relevance for Chronic Diseases	Comment, Advantage/Disadvantage
EPR	Original ROS and RNS measurements	Whole blood, cells, cellular compartments	Very limited, fresh samples must be used	EPR spectroscopy, spin probe needs to be added to stabilize electron spins	Proposed and used in some neurological disease such as human prion diseases [151]. In small-scale study with 100 middle-aged subjects, capillary blood was measured by EPR. ROS increased with age, EPR measures were well correlated with PCs and TBARS [152].	Measures cause and original OS species, techniques rarely available
8-OH-dhG	RNA/DNA oxidative breakdown products	Urine, plasma/ serum	2 y at −80 °C [19]	ELISA, chromatography	Meta-analysis related 8-OH-dhG with CVD [18] and solid tumors [153].	Not recommended by EFSA alone but as support measure [150]
DNA double strand break, COMET assay	DNA oxidative breakdown	Viable cells, e.g., leukocytes	Fine as long as cells can be kept viable or fresh samples are used	Electrophoresis with fluorescence microscopy. Modifications with endonuclease III treatment for damaged pyrimidine base detection and formamidopyrimidine DNA glycosylase for purine bases	Higher level of DNA damage in smokers vs. non-smokers in meta-analysis of 38 studies (SMD = 0.55 [154]). Increased short-term DNA damage after exercise in a meta-analysis with 35 studies [155].	Modified method accepted by EFSA [150]
MDA	Lipid oxidation product	Plasma/ serum	<3 weeks at −20 °C [156]. Doubling of concentration when stored on ice for 36 h	Spectrophotometrically, chromatography	Meta-analysis showed MDA elevation in patients with PD [17], AMD [157], and COPD [158]. In T2D subjects receiving vitamin C or E, TBARS/MDA was sign. reduced in plasma [159].	Not recognized by EFSA [150]
F2-isoprostanes	Lipid oxidation product	Plasma/serum or urine	6 months at −80 °C (plasma), changes with freezing/thawing cycles [160,161]	GC-MS, LC-MS, ELISA. ELISA yielding often >50% higher levels due to cross-reactivity [162]. Urine levels 20–100 times higher than plasma; total F2-isoP or esterified ones are superior to free F2-isoP as the latter can be formed independently from ox. stress [163].	Meta-analysis found strong increases of F2-isoprostanes for respiratory and kidney diseases, weaker ones for metS, smoking, and hypertension [163]. Relations in meta-analyses between cardiac arrest and F2-isoprostanes [164], positive airway pressure treatment in subjects with sleep apnoea [165] and depression [166].	Recognized by EFSA if detected by MS-MS [150]
Phosphatidyl-choline hydroperoxides (PCOOH)	Lipid peroxidation	Plasma/serum	Sample stability (plasma) at −78 °C was reported, but increased levels at 4 °C and −20 °C [167]	LC-CL or LC-MS.	Increased PCOOH concentrations with ageing [168] and in dialysis patients with diabetic nephropathy [169].	Recognized by EFSA [150]
oxLDL	Lipid oxidation	Plasma/serum	Whole blood samples 36 h on ice [170]. For plasma, several months at -20 °C [171], anti-freezing recommended	Antibodies	Meta analyses showing positive association between oxLDL and atherosclerotic CVD [23], sleep apnoea [172], and reducing oxLDL by olive oil [173], systematic review of oxLDL Ab and CAD.	Recognized by EFSA [150]
AOPPs	Protein oxidation, often of albumin	Plasma/serum	At least 6 months at −20 °C [174]	Commercial colorimetric assays	Relation to coronary syndrome [175], Crohn’s disease [176], diabetes [177].	Results from small and medium-scale studies
Protein carbonyls (PCs, subgroup of AOPPs)	Protein oxidation	Plasma/serum	At least 1 month at −80 °C [178]	Spectrophotometry, antibodies, or MS/proteomics, mostly following derivatization with DNP [179]	Small-scale clinical studies relating PCs to Alzheimers disease [180] and T2D treatment [181], among others.	Only results from small-scale disease related studies
Nitrated albumin	Protein oxidation	Plasma/serum	Nitrated proteomics standard UPS1 stable for 9 months at −20 °C	Nitrated albumin	Protein oxidation.	Plasma/serum
γH2AX	Histone modification	In cells (leukocytes etc.)	Unknown	Immunological (ELISA), combined with e.g., flow-cytometry	Mostly been used as a biomarker for cancer [182,183].	Unknown if suitable marker of ox. stress, rather marker of apoptosis [184]
GPx, SOD, CAT	Antioxidant enzymes	Serum (SOD3, GPX3), erythrocytes (CAT)	Stable when stored on ice for 48 h [185] or frozen at −80 °C for 21 months [185]	Spectrophotometrically	Lower levels of SOD, CAT and GPx in meta-analysis with subjects with CHD [186] and with PD [187]. Not recognized by EFSA [150].	Marker of antioxidants, not necessarily pro-oxidants.
Total antioxidant capacity	Antioxidant capacity	Serum, plasma	Depending on stability of plasma constituents albumin, phenolics, uric acid, tocopherols etc. Likely several months at −80 °C	Spectrophotometric	Sign. lower RR for all-cause mortality for subjects with higher FRAP and TRAP status, though only based on 3 and 2 observational studies, resp. [188]. In hemodialysis patients, sign. lower antiox. Capacity found vs. healthy controls [189]. In T2D subjects, those with more complications showed higher levels of TAC (determined by FRAP, [190]).	Only marker of antioxidants, not necessarily pro-oxidants
Nrf2	Antioxidant transcription factor	Leukocytes, tissue	Possibly several months at −80 °C (protein extract), [191]	Transcriptomics, immunologic (ELISA), preferably in nuclear fraction	High expression of Nrft in meta-analysis (17 studies) of patients with solid tumors [192]. Overexpression of Nrf2 in meta-analysis of AD and PD, though ARE related genes (*n* = 31) were down-regulated, likely due to MAFF overexpression [193].	Difficult to interpret data, time and tissue dependency of measures, requiring isolation of nuclear fraction
Index—thiol ratio	Index—antioxidant/oxidant ratio test	Plasma, serum	At least 6 months at −80 °C [194]	Spectrophotometric	Increased ratio of disulfide to native thiols related to CVD [195], respiratory diseases and metabolic disorders, [27], as well as cancer, ageing, and neurodegenerative diseases [196], and obesity [197].	Interesting marker and physiologically meaningful, lack of data
Index—GSH:GSSG ratio	Index—antioxidant/oxidant ratio test	Intracellular fractions	Unclear, challenge to isolate fraction as very unstable [198]	Spectrophotometric (commercial assays available)	No meta-analysis. Clinical studies employing GSH:GSSG ratio reviewed earlier [27], listing 2 dozen individual studies, mostly small-scale. Some studies reporting on blood pressure, virus-related respiratory problems, mercury exposure and overweight etc. were related to a higher oxidized ratio of GSH:GSSG.	Delicate marker as analytically challenging
DAQS (Dietary Antioxidant Quality Score)	Dietary index	Diet	Calculates sum of antioxidants, creates a score, considering recommended intakes. Includes vitamin A, C, E; Se, Mn and Zn	FFQ	Employed in women, finding a significant relation between higher antioxidant intake and higher bone-mineral density [199]. A modified version, assigning scores of 0 or 1 for food items was developed [200], and a significant interaction with polymorphisms of gene region on chromosome 6P21 was found.	Only including dietary aspects, not further host-factors. Disregards secondary plant compounds
CDAI (Composite Dietary Antioxidant Index)	Dietary index	Diet	Ranks antiox. intake vs. population norms. Included also vitamins A, C, E, and minerals Se, Mn, Zn	FFQ	Original approach by Wright et al. included carotenoids, flavonoids, vitamin E, C and Se, i.e., also phytochemicals [201]. In their study, a higher index was related to lower lung cancer risk.	Only including dietary aspects, not further host-factors

AD: Alzheimer’s disease, CAT: catalase, CHD: coronary heart disease, CVD: cardio-vascular disease, EFSA: European Food Safety Authority, EPR: electron paramagnetic resonance, FRAP: ferric reducing antioxidant power assay, GPx: glutathione peroxidase, GSH:GSSG: reduced vs. oxidized glutathione, LC: liquid chromatography, MS: mass spectrometry, Nrf2: nuclear factor erythroid-2 related factor 2, PCOOH: phosphatidyl-choline hydro-peroxides, PD: Parkinson’s disease, SOD: superoxide dismutase, TAC: total antioxidant capacity, T2D: type-2 diabetes, γH2AX: phosphorylated H2A histone family member X.

**Table 2 antioxidants-10-00414-t002:** Overview of common markers of inflammation—advantages and disadvantages.

Marker	Class of Marker	Measured in: Matrix	Stability	Techniques Used	Relevance for Chronic Diseases	Comments, Advantages/Disadvantages
Cytokines: IL1β, IL-6, IL-8, IFγ, TNF-α,	Pro-inflammatory cytokines	Plasma, serum, or expression of mRNA in tissue	Stable for 3 freeze-thaw cycles reported for IL-6, IL-10, IFNγ, and IL-2 [309]. Stable during 1 y at −80 °C (IL-1β, IFγ, IL-6, TNF-α) [310]	ELISA, multiplex bead array assays [311], mRNA levels by PCR	IL-6: In meta-analyses related to cognitive decline in elderly without dementia [312], more severe COVID-19 complications [313], IBS [314], polymorphisms of IL-6 related to lower risk of obesity [315].	Primary markers of inflammation, low half-lives in blood (minutes)
					IL-8: Meta-analyses shown relations between elevated IL-8 and lupus erythematodes [316], hepato-cellular carcinoma [317], periodontitis [318], but not obesity [319].	
					IFγ: Polymorphisms related in meta-analysis of 9 case-control studies to breast cancer [320], in a meta-analysis of 8 studies on cervical cancer [321]. No sign. association with mycoplasma caused pneumonia in meta-analysis of 6 small-medium scale studies [322].	
					TNF-α: higher circulating conc. in individuals with obesity [323]. Meta-analysis of 5 prospective studies failed to show relation with T2D RR, unlike IL-6 and CRP [324]. Increased TNF-α (and IL-6 and 8) in breast-cancer based on meta-analysis [325].	
Leukocyte count	Cell counting	Whole blood	Cell counting should be done on fresh whole blood samples, within 3 h [326]. Freezing for 15 d at −70 °C resulted in sign. changes in cell counts [327]	In clinic, several automated systems, including impedance and optical systems, as well as image cytometers [328]	Increased leukocyte count (>11 × 10^9^/L) associated with worse outcome in stable CAD and even in the general population [329], in a meta-analysis related with MetS [330] and also with T2D [331].	Interesting proxy-marker of inflammation, as leukocytes are major secreters of cytokines
Lymphocyte count					Low lymphocyte count found in meta-analyses of COVID-19 [332], also used in HIV as decision-helper for starting anti-viral therapy [333].	
Eosiniphil counts					Normally <0.5 × 10^9^ cells/L. Eosinophils increased in allergic diseases, e.g., asthma as shown in a meta-analysis [334] and also in autoimmune diseases, such as celiac disease [335], and in IBD (both CD and UC) [336]. Elevated also in Hodgkin’s lymphoma [337].	
Monocyte counts					Monocytosis indicated by >0.8 × 10^9^/L in adults. Frequently associated with infections, e.g., malaria [338], autoimmune diseases such as rheumatoid arthritis [339], IBD (1020), colorectal cancer as shown in a meta-analysis [340], obesity [341], and schizophrenia as seen in a meta-analysis [342]; no elevation found in meta-analysis of T2D [331].	
Basolphil counts					Number normally <0.20 × 10^9^/L [343]. Elevated level (>2.0 × 10^9^/L) associated with autoimmune inflammation and allergies [344]. Low level of basophils associated with rheumatoid arthritis [345].	
Neutrophil-lymphocyte ratio (NLR)					High NLR reported for poor glycemic control in T2D [346]. Multiple studies showed that increased NLR was associated with poor prognosis in a variety of cancers, such as colon, pancreatic, stomach, and lung cancer. In meta-analysis of >100 studies and 40,000 patients, an NLR >4 was associated with worse solid tumor survival rates [347].	
Lymphocyte –monocyte ratio (LMR)					LMR associated with stroke severity and poor outcome, also according to meta-analysis [348]. Low LMR was sign. associated in a meta-analysis with lower overall survival in patients with ovarian cancer [349] and with solid tumors due to meta-analysis of 29 studies [350].	
SII					In meta-analysis of 22 articles and >7500 patients, SII proposed as good predictor for tumor progression and survival in several cancers [351].	
CRP	Acute phase proteins/reactants	Blood, CRF,	11 y at −80 °C [352]	ELISA and other immune-techniques [353]	Meta-analyses showed higher levels in blood of PD [354], CHD and stroke mortality [355]. Increased tyrosine-nitrated levels in subjects with CAD [356] and with hemodialysis vs. controls [357].	Very frequently used marker
SAA		Serum, plasma	At least 17 d at −20 °C [358], no report on storability at −80 °C	In clinical practice automated latex agglutination immunoassay [359]	Strong relation between elevated SAA and obesity [14], coronary heart disease [15], poor overall survival in individuals with solid tumors [360] and worse outcomes in lung cancer [361] in meta-analysis.	Very frequently used marker
Haptoglobin		Serum, plasma	Stability of haptoglobin in bovine plasma at least 3 weeks at −20 °C and −80 °C [362] and 120 d in saliva at −20 °C [363]	Nephelometric methods in plasma on different automated laboratory systems [364], spectrophotometry, immunological methods, gel electrophorosis	In a recent meta-analysis, the haptoglobin genotype was associated with cardiovascular outcomes [365]. Haptoglobin also increases in inflammatory diseases, in cigarette smokers, during nephrotic syndrome [366] and in rheumatoid arthritis [367]	Interesting though less frequently employed marker
Mannose-binding lectin (MBL)		Serum, plasma	Pure protein can be stored for 1 y at −80 °C [368], no data on plasma.	ELISA	In meta-analysis of MBL and polymorphisms, significant associations were found to sepsis [369], and vulvovaginal infections [370], but not to hepatitis B infection risk (though severity) [371]	Interesting but less frequently employed marker
Acute phase reactant -procalcitonin		Serum, plasma	Small losses of 10% with several year storage at −80 °C [372]	ELISA	Meta-analysis of pneumonia [373] did not suggest that marker was useful to differentiate between viral and bacterial diseases, while it allowed to differentiate in another meta-analysis between infection and IBD [374]	Marker of bacterial, not viral infection or general inflammation marker
ICAM-1, VCAM-1	Endothelial marker	Soluble forms in serum, plasma	Some months in lyophilized form in kits at −20 °C, no published data on stability in samples	ELISA	In meta-analysis of 15 prospective studies with T2D, increased ICAM-1 but not VCAM-1 related to higher risk of T2D [375]. In meta-analysis of 18 case-control studies of cancer risk, ICAM-1 polymorphism was related to cancer risk [376], similar to meta-analysis of 12 case-control studies on T2D [377] and CHD in meta-analysis of 11 case-control studies [378].VCAM-1, but not ICAM-1 associated with preeclampsia in a meta-analysis of 21 studies [379].	Marker of endothelial health, rather not general inflammation
PGE2	COX-2 related marker	Serum, plasma	Unclear, but likely short, even at −80 °C	ELISA, mRNA measurement, MS-MS.	Elevated PGE2 levels in women with obesity [380], cancer, including colorectal cancer [381] and ovarian cancer [382], T2D patients [383], contribution to neurological disorders assumed [384].	As stimulated by cytokines interesting marker of systemic inflammation
NF-κB	Pro-inflammatory transcription factor	Leukocytes, body tissue	2 months at −80 °C in isolated protein extract [385]	ELISA, transcriptomics. Extraction of nuclear fraction advised.	In meta-analysis, increased expression of NF-kB in individuals with worse solid tumor outcomes [25], both in cytosol and nucleus. Increased expression in meta-analysis of non-small cell lung cancer [386], but only when determined in nuclear fraction. Isolated mycocytes from subjects with obesity showed elevated NF-kB expression [387].	Difficult to interpret, time and tissue dependency of measures
Neopterin	Macrophage product	Plasma, urine	3 d at RT, 3 months at −20 °C [111]	Fluorescence, HPLC, ELISA	Plasma levels were related to homocysteine and risk of CVD [388], and as well as to cancer such as oviaran cancer [389]. It was also shown to be modulated by age, gender, and BMI [390].	Rather marker of immune cell activation
GlycA	Composite marker of glycosylation	Plasma, serum	Unclear, glycosy-lation increases protein stability [391], some months at −80 °C assumed	NMR	Strong relation to total mortality and cancer mortality [392], exercise and visceral fat [393]	Promising novel marker, expensive equipment needed (NMR)
HDL inflammatory marker (HII)	Lipoprotein marker	Plasma, serum	Based on stability of HDL in plasma, approx. 1–4 weeks at −80 °C, with cryo-preservants possibly longer [394]	Oxidation of LDL standard and protection by HDL fraction from individuals, dichloro-fluorescein fluorescence measurement	Higher HII associated with sepsis and shock in individuals with organ failure [395], poor outcome in subjects with hemodialysis [396] and diabetes [397].	Analytically challenging
Dietary inflammatory index (DII)	Dietary marker	dietary intake	n.a.	FFQ	Higher DII related in meta-analyses to mortality of cancer, CVD, and total mortality [398], obesity [399] and cancer [400]	Only including dietary aspects, not further host-factors

AD: Alzheimer’s disease, COX-2: cyclo-oxygenase 2, CRP: c-reactive protein, CVD: cardiovascular disease, ELISA: Enzyme-linked immune assay, FFQ: food frequency questionnaire, HDL: high density lipoproteins, IBD: inflammatory bowel diseases, ICAM-1: intercellular adhesion molecule 1, na: not applicable, NF-κB: nuclear factor kappa B, NMR: nuclear magnetic resonance, PCR: polymerase-chain reaction, PD: Parkinson’s disease, PGE2: prostaglandin E2, SAA: serum amyloid alpha, T2D: type 2 diabetes, VCAM-1: vascular adhesion molecule 1.

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
