# Peer review of "Common and Novel Markers for Measuring Inflammation and Oxidative Stress Ex Vivo in Research and Clinical Practice—Which to Use Regarding Disease Outcomes?"

_antioxidants, 2021, doi:10.3390/antiox10030414_

Round 1
Reviewer 1 Report
The aim of this review is “to provide an overview of markers of inflammation and OS which are prominently utilized in research and in clinical practice ……” (l 147).
This aim is achieved: the review is comprehensive and is likely to become an Introduction of new comers to the field.
Unfortunately, the authors contribute to the knowledge of the reader but not much to the understanding of the reviewed knowledge. My recommendation to the authors is to add a chapter to the review with analysis of the information.
Specifically, it appears that in some cases, but not always, the OS, as determined on the basis of different biomarkers correlate with each other. The same is probably true for markers of inflammation.. Is the available data sufficient to look for correlations between markers of OS and markers of inflammation ? Any answer to these questions is meaningful.. Experimental details of different published studies,, particularly the time of measurements in ex-vivo studies , may yield information on the role of OS in inflammation.
Author Response
Reviewer 1
The aim of this review is “to provide an overview of markers of inflammation and OS which are prominently utilized in research and in clinical practice ……” (l 147).
This aim is achieved: the review is comprehensive and is likely to become an Introduction of new comers to the field. Unfortunately, the authors contribute to the knowledge of the reader but not much to the understanding of the reviewed knowledge. My recommendation to the authors is to add a chapter to the review with analysis of the information.
Reply : We have tried to improve the understanding by adding further interpretation to the individual chapters, to directly provide more information when reporting clinical examples. In addition, the conclusive chapter is giving now some further guidelines for the choice of markers, specifically, a summary of recommendations has been added, please see lines 1361 ff.
Specifically, it appears that in some cases, but not always, the OS, as determined on the basis of different biomarkers correlate with each other. The same is probably true for markers of inflammation.. Is the available data sufficient to look for correlations between markers of OS and markers of inflammation ? Any answer to these questions is meaningful.
Reply : This is indeed a very interesting aspect. We have added a section in the conclusions to insert some information on oxidative stress correlates, inflammation correlates, and also regarding the correlation of oxidative stress and inflammation, please see lines 1261 ff. We kept this somewhat succinct to not add too much length to the already long manuscript.
Experimental details of different published studies, particularly the time of measurements in ex-vivo studies, may yield information on the role of OS in inflammation.
Reply : Details on time of measurements, when available, were also added, see in the individual chapters, as well as in the summary/conclusions a paragraph was added, please see lines 1289 ff. Also a figure on this was inserted, please see page 36.
Reviewer 2 Report
General comments:
The topic of the manuscript is an overview of current research on the effects of oxidative stress on the human organism and presents methods for determination of the level of oxidative stress based on various types of indicators. This is extremely important, since the disturbance of the antioxidant balance in the organism is associated with occurrence of a number of chronic diseases. Timely diagnosis of the levels of oxidants and antioxidants in the organism may facilitate a quick response and treatment. Therefore, I believe that the study is in the scope of Antioxidants.
The Authors thoroughly discussed the subject and fully achieved the assumed aim of the study. Simultaneously, they emphasized that there is no single universal antioxidant stress index characteristic of individual diseases, and showed indicators that definitely suggest disturbances in the antioxidant balance of the organism. More than one indicator should be taken into account while assessing this condition.
Specific comments:
- Chapter 4. – please explain the EFSA acronym
- References: I believe that relevant literature was used in the study to explore the issues fully.
Comments on the References:
Reference 259 – please complete the information
References 266, 302, 380, 393, 407, 413 and other – please present the name of the journal as required by editorial guidelines
References 288 and other – follow the editorial guidelines
Reference 304 – please verify the title of the publication
Author Response
Reviewer 2
General comments:
The topic of the manuscript is an overview of current research on the effects of oxidative stress on the human organism and presents methods for determination of the level of oxidative stress based on various types of indicators. This is extremely important, since the disturbance of the antioxidant balance in the organism is associated with occurrence of a number of chronic diseases. Timely diagnosis of the levels of oxidants and antioxidants in the organism may facilitate a quick response and treatment. Therefore, I believe that the study is in the scope of Antioxidants.
The Authors thoroughly discussed the subject and fully achieved the assumed aim of the study. Simultaneously, they emphasized that there is no single universal antioxidant stress index characteristic of individual diseases, and showed indicators that definitely suggest disturbances in the antioxidant balance of the organism. More than one indicator should be taken into account while assessing this condition.
Reply: We appreciate the reviewer’s constructive comments. We have now also further highlighted, in an additional chapter in the conclusions, that more than 1 marker should be assessed, also emphasizing the correlation between various markers, please see lines 1169ff.
Specific comments:
- Chapter 4. – please explain the EFSA acronym
Reply : This was done as suggested, please see line 463.
- References: I believe that relevant literature was used in the study to explore the issues fully.
Reply : We appreciate this comment.
Comments on the References:
Reference 259 – please complete the information
Reply : We have updated the reference.
References 266, 302, 380, 393, 407, 413 and other – please present the name of the journal as required by editorial guidelines
Reply : We have added this information and apologize for the omission.
References 288 and other – follow the editorial guidelines
Reply : We have updated the reference.
Reference 304 – please verify the title of the publication
Reply : We have updated the reference accordingly.
Reviewer 3 Report
- The manuscript entlited: “Common and Novel Markers for Measuring Inflammation and Oxidative Stress ex vivo in Research and Clinical Practice – which to use Regarding Chronic Disease Outcomes ?” authored by Alain Menzel colleagues, is very relevant and interesting from a scientific point of view, however some small corrections should be made.
- Many chronic diseases modify the antioxidant balance in the human organism and it’s very important to act quickly with a correct diagnosis. This topic introduce an overview both the effects of oxidative stress on the body and various parameters to measure it.
- The authors explored the topic and they obtained the purpose of the study. They reported a large array of markers related to oxidative stress and inflammation and they showed how it is not enough to consider just one parameter to evaluate alteration of the antioxidant balance but it’s necessary to chhose the most appropriate one.
- The paper is well written and text is clear to read.
- The conclusions are consistent with the evidence and arguments presented
Specific indications:
- The following acronyms should be reported in extenso at their first appearence in the text
Chapter 2. PUFAs
Chapter 3. EPA, DHA, WBC and NET
- About references:
Reference 29 – please complete the information
Reference 303, 304– please verify the title of the publication
References 5, 14, 15, 17, 18, 19, 23 and other – please present the right number of authors before putting et al. as required by editorial guidelines
Author Response
Reviewer 3
- The manuscript entlited: “Common and Novel Markers for Measuring Inflammation and Oxidative Stress ex vivo in Research and Clinical Practice – which to use Regarding Chronic Disease Outcomes ?” authored by Alain Menzel colleagues, is very relevant and interesting from a scientific point of view, however some small corrections should be made.
Many chronic diseases modify the antioxidant balance in the human organism and it’s very important to act quickly with a correct diagnosis. This topic introduce an overview both the effects of oxidative stress on the body and various parameters to measure it.
The authors explored the topic and they obtained the purpose of the study. They reported a large array of markers related to oxidative stress and inflammation and they showed how it is not enough to consider just one parameter to evaluate alteration of the antioxidant balance but it’s necessary to chose the most appropriate one.
Reply : We agree with this succinct summary.
- The paper is well written and text is clear to read.
The conclusions are consistent with the evidence and arguments presented
Reply : We appreciate the comment of the reviewer.
Specific indications:
- The following acronyms should be reported in extenso at their first appearence in the text
Chapter 2. PUFAs
Chapter 3. EPA, DHA, WBC and NET
Reply : We have inserted the explanation when mentioning these the first time, in addition, they can be found in the list of abbreviations.
2. About references:
Reference 29 – please complete the information
Reply : We have complete the information in the reference.
Reference 303, 304– please verify the title of the publication
Reply : The references have been updated.
References 5, 14, 15, 17, 18, 19, 23 and other – please present the right number of authors before putting et al. as required by editorial guidelines
Reply : It has been corrected, we apologize for the mistake.